# The cholesterol biosynthesis pathway regulates IL-10 expression in human Th1 cells

Esperanza Perucha [1,2], Rossella Melchiotti[3], Jack A Bibby[1,2], Wing Wu[1,2], Klaus Stensgaard Frederiksen [4], Ceri A. Roberts[2,14], Zoe Hall[5], Gaelle LeFriec[6], Kevin A. Robertson[7], Paul Lavender[8], Jens Gammeltoft Gerwien[4,15], Leonie S. Taams [2], Julian L. Griffin[5], Emanuele de Rinaldis[3], Lisa G.M. van Baarsen[9,10], Claudia Kemper[6,11,12], Peter Ghazal[7,13] & Andrew P. Cope [1,2]

The mechanisms controlling CD4$^+$ T cell switching from an effector to an anti-inflammatory (IL-10$^+$) phenotype play an important role in the persistence of chronic inflammatory diseases. Here, we identify the cholesterol biosynthesis pathway as a key regulator of this process. Pathway analysis of cultured cytokine-producing human T cells reveals a significant association between IL-10 and cholesterol metabolism gene expression. Inhibition of the cholesterol biosynthesis pathway with atorvastatin or 25-hydroxycholesterol during switching from IFNγ$^+$ to IL-10$^+$ shows a specific block in immune resolution, defined as a significant decrease in IL-10 expression. Mechanistically, the master transcriptional regulator of *IL10* in T cells, c-Maf, is significantly decreased by physiological levels of 25-hydroxycholesterol. Strikingly, progression to rheumatoid arthritis is associated with altered expression of cholesterol biosynthesis genes in synovial biopsies of predisposed individuals. Our data reveal a link between sterol metabolism and the regulation of the anti-inflammatory response in human CD4$^+$ T cells.

[1] Academic Department of Rheumatology, King's College London, London SE1 1UL, UK. [2] Department of Inflammation Biology, School of Immunology and Microbial Sciences, Centre for Inflammation Biology and Cancer Immunology, King's College London, London SE1 1UL, UK. [3] National Institute for Health Research Biomedical Research Centre, Guy's and St Thomas' NHS Foundation Trust and King's College London, London SE1 9RT, UK. [4] Global Drug Discovery, Novo Nordisk A/S, 2880 Bagsvaerd, Denmark. [5] Department of Biochemistry and the Cambridge Systems Biology Centre, University of Cambridge, Cambridge CB2 1QW, UK. [6] MRC Centre for Transplantation, King's College London, London SE1 9RT, UK. [7] Division of Infection and Pathway Medicine, University of Edinburgh, Edinburgh EH16 4SB, UK. [8] School of Immunology and Microbial Sciences, King's College London, London SE1 9RT, UK. [9] Amsterdam Rheumatology and immunology Center (ARC), Department of Rheumatology and Clinical Immunology, Amsterdam UMC, University of Amsterdam, 1105 AZ Amsterdam, Netherlands. [10] Department of Experimental Immunology, Amsterdam UMC, University of Amsterdam, 1105 AZ Amsterdam, Netherlands. [11] Laboratory of Molecular Immunology and the Immunology Center, National Heart, Lung, and Blood Institute (NHLBI), National Institutes of Health (NIH), Bethesda, MD 20892, USA. [12] Institute for Systemic Inflammation Research, University of Lübeck, 23562 Lübeck, Germany. [13] Systems Immunity Research Institute, Medical School, University of Cardiff, Cardiff CF14 4XN, UK. [14] Present address: Cellular and Molecular Therapy, NHS Blood and Transplant, Bristol BS34 7QH, UK. [15] Present address: Rheumatology NEC, Eli Lilly, 2730 Copenhagen, Denmark. Correspondence and requests for materials should be addressed to E.P. (email: esperanza.perucha@kcl.ac.uk) or to A.P.C. (email: andrew.cope@kcl.ac.uk)

CD4+ T-helper (Th) effector cells are integral to the immune response, differentiating into Th1, Th2 and Th17 subsets tuned to respond to a wide range of pathogens and environmental insults[1,2]. Th1 cells produce the signature cytokine interferon-γ (IFNγ) that functions to efficiently eradicate intracellular pathogens. While defects in the IFNγ pathway lead to uncontrolled infection[3,4], Th1 responses must be tightly controlled to prevent host tissue damage following pathogen elimination. The restoration of immune homeostasis can be defined by the expression of interleukin-10 (IL-10), a prototypic anti-inflammatory cytokine that orchestrates termination of immune responses[2,5–7]. The absence of this regulatory checkpoint may lead to persistent inflammatory responses, while uncontrolled expression of IL-10 may impede eradication of infectious organisms[8,9]. Despite its importance, our understanding of the molecular switches that control how CD4+ T cells acquire the capacity to produce IL-10 remains incomplete. Cytokines such as IL-12, IL-27 or type I IFN in combination with T cell receptor and co-stimulatory receptor engagement have been shown to induce IL-10[10–12]. These signals are propagated via downstream signalling intermediates (extracellular signal-regulated kinase (ERK), nuclear factor for activated T cells (NFAT) and nuclear factor-κB (NF-κB)) and induce expression of c-Maf, a master regulator of *IL10* in T cells and, together with other transcription factors such as IRF4, AhR or Blimp-1, activate the transcription of *IL10*[10,11,13–17]. These studies indicate a high level of complexity in IL-10 regulation.

We have developed in vitro models of human T cell activation and differentiation that have allowed us to elucidate more precisely the pathways that promote IL-10 expression. In humans, a range of studies have provided evidence that the autocrine engagement of CD46 by complement is critically important for IFNγ secretion and Th1 induction, and, together with IL-2 receptor signalling, induces a switch to IL-10 co-production in Th1 cells with a transition into a self-regulatory, contracting phase[5]. Accordingly, patients deficient in CD46 cannot generate Th1 responses and suffer from recurrent infections[18,19], while dysregulation in this CD46/IL-2R crosstalk leads to a reduced IL-10 switching and hyperactive Th1 responses in rheumatoid arthritis (RA) and multiple sclerosis[18,20].

To meet the energy demands required for T cell effector function, T cell activation is accompanied by a metabolic shift from oxidative phosphorylation to aerobic glycolysis, allowing increased metabolic flux towards anabolic pathways in order to provide key metabolites required for the acquisition of effector function[21]. This activation-induced metabolic reprograming is coordinated by canonical metabolic signals, such as mammalian target of rapamycin, and immune specific cytokines and transcription factors[21,22], as well as the complement receptor CD46[22], and has been extensively studied in recent years[23]. Notably, this metabolic reprogramming also involves changes in the cholesterol biosynthesis pathway (CBP)[24], although much less is known about its involvement in the regulation of immune function. In keeping with a metabolic requirement for T cell activation, signalling via the T cell receptor enhances cholesterol metabolism by increasing the transcription of key enzymes of the CBP[25]. Moreover, alterations in the expression of liver X receptor-β and sterol regulatory element-binding protein 2 (SREBP-2), which fundamentally perturbs the integrity of cholesterol biosynthesis, have a direct impact on T cell proliferation and effector functions[25,26], while inhibition of mevalonate metabolism in T-regulatory cells (Tregs) has been shown to block their suppressive capacity[27]. In addition to this, the CBP generates not only cholesterol, an essential building block of cellular membranes, but it also provides metabolites with known roles in cellular and immunological processes. For example, the pathway generates non-sterol mevalonate derivatives such as farnesol and geranylgeraniol, implicated in the post-transcriptional prenylation of intracellular proteins such as Rho and Ras[24]. It also synthesises 7-dihydrocholesterol, the precursor of immunomodulatory vitamin D3, which has been directly implicated in IL-10 production by CD4+ T cells[28]. The CBP also generates oxysterols, derivatives of cholesterol such as 25-hydroxycholesterol (25-HC), that is part of the type I IFN anti-viral response[29,30], and 7-ketocholesterol, a natural ligand for the transcription factor AhR[31], which is also implicated in IL-10 gene expression in Tr1-like cells[15].

In the present study, we demonstrate that targeted perturbation of the CBP in human CD4+ T cells blocks the switch from effector (IFNγ+) to regulatory (IL-10+) function and has a direct and highly specific impact on IL-10 messenger RNA (mRNA) and protein levels. Furthermore, cholesterol pathway blockade downregulates the expression of the transcription factor c-Maf. Moreover, transcript levels of key enzymes belonging to the CBP are significantly related to disease progression in subjects at high risk of developing RA, providing an in vivo correlation to our model. Our work provides the first evidence of lipid metabolism regulating the contraction of the human CD4+ T cell response and offers new insights into the complex relationship between lipid metabolism, inflammatory disease and host immunity.

## Results

**Th1 switching to IL-10 is linked to cholesterol biosynthesis**. Upon activation with α-CD3 and α-CD46, CD4+ Th1 cells undergo a self-regulatory cycle during which IFNγ+ cells co-express IL-10, followed by an IL-10 single-positive regulatory phase, creating a Th1 switching cycle[18]. To unravel the molecular mechanisms controlling the expression of IL-10 during Th1 switching, we stimulated human peripheral blood CD4+ T cells with α-CD3 and α-CD46 for 36 h to generate cells at different phases of the Th1 switching cycle, as previously described[18]. Double-negative IFNγ−IL-10−, single-positive IFNγ+IL-10− and IFNγ-IL-10+ cells and double-positive IFNγ+IL-10+ subsets were purified by cell sorting and subjected to gene expression profiling (Supplementary Figure 1a and b). Initial clustering analysis revealed that populations did not segregate based on their gene expression (Supplementary Figure 1c). We reasoned that genes involved in the regulation of IL-10 expression would be present in the pre-IL-10 expression phases of the cycle (IFNγ−IL-10−, IFNγ+IL-10− subsets, black gates, Fig. 1a) and that their expression might correlate with IL-10 protein levels in the double-positive population (blue gate, Fig. 1a). To test this, datasets were interrogated for genes present in either the double-negative or IFNγ+IL-10− subsets whose expression correlated with IL-10 expression in the double-positive population. Levels of IL-10 protein were quantified by flow cytometry using median fluorescence intensity (MFI) as an indicator of the number of IL-10 molecules per cell, and normalised to the MFI value of the double-negative population to account for inter-experimental variability. Using this approach we identified 111 genes expressed in both the double-negative and single-positive IFNγ+IL-10− populations correlating inversely (top) or directly (bottom) with IL-10 levels (Fig. 1b and Supplementary Data 1). From these gene sets, Ingenuity Pathway Analysis (IPA) ranked multiple pathways linked to cholesterol biosynthesis as being the most significantly associated with IL-10 expression (Fig. 1c). IPA analysis also revealed an inverse relationship between genes belonging to the CBP expressed by the pre-IL-10 populations and IL-10 protein levels in the double-positive IFNγ+IL-10+ population, suggesting that active cholesterol biosynthesis might be required to license Th1 cells to express IL-10. This signal was further validated by quantitative real-time

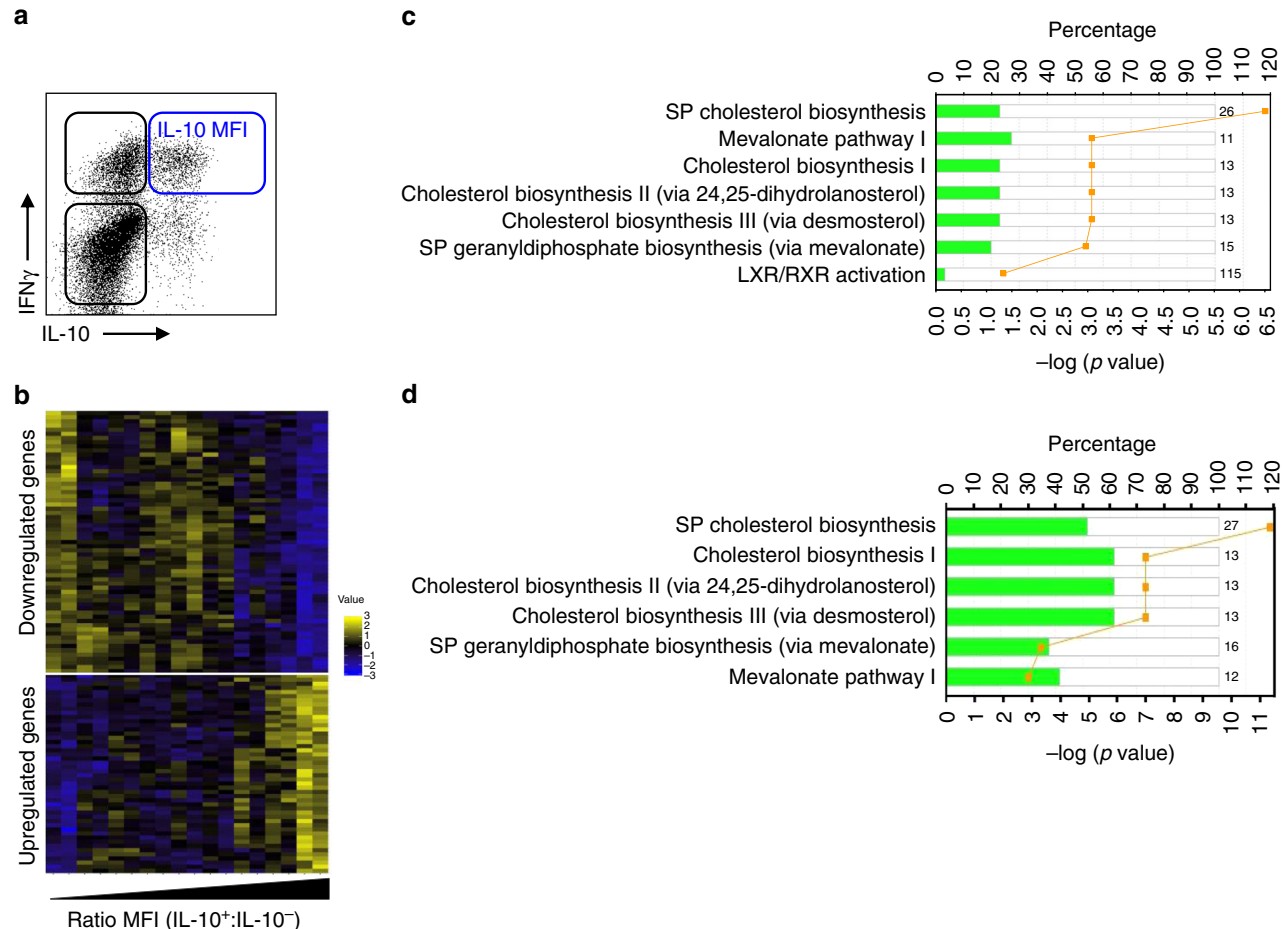

**Fig. 1** Metabolic regulation of T-helper type 1 (Th1) switching in primary human CD4+ T cells. **a** Schematic flow cytometric dot-plot of the life cycle of human Th1 switching; double-positive cells are highlighted with a blue gate, while IL-10-negative populations are shown within a black gate; **b** Heatmap of 111 significantly expressed genes in the double-negative (IFNγ−IL-10−) and single-positive (IFNγ+IL-10−) populations associated with changes in IL-10 protein expression, as measured by median fluorescence intensity ratio (MFI+:MFI−) for IL-10 expression in the double-positive (IFNγ+IL-10+) population. **c** Ingenuity Pathway Analysis (IPA) based on genes from **b** (SP: superpathway). Bars represent the percentage of the number of genes mapping to a particular pathway, coloured in green to note the inverse correlation. The number indicates the number of total genes ascribed to the pathway. The yellow line represents the P value as calculated by Fisher's test and corrected for multiple testing using the Benjamini–Hochberg correction. **d** IPA based on genes differentially expressed between CYT-1- and CYT-2-expressing Jurkat T cells and analysed and annotated as in **c**

PCR (qPCR) for selected genes within the CBP (Supplementary Figure 1d).

CD46 signals through one of two intracellular cytoplasmic tails: CYT-1 promotes Th1 IFNγ expression, while CYT-2 promotes IL-10 switching[18]. To further investigate the link between cholesterol biosynthesis and IL-10 expression, we compared the transcriptome of Jurkat T cells stably expressing CYT-1 or CYT-2; the transcriptome of untransduced Jurkat T cells was used as control. Principal component analysis (PCA) identified three distinct subpopulations (Supplementary Figure 1e), indicating that signalling through either CYT-1 or CYT-2 tails was sufficient to drive distinct transcriptional profiles. Once again, IPA of differentially expressed genes identified cholesterol biosynthesis and related biosynthetic pathways (mevalonate and geranyldiphosphate) as highly enriched (Fig. 1d). Moreover, and as observed in Th1 switching primary CD4+ T cells, these genes were downregulated in Jurkat T cells expressing CYT-1 (effector) when compared to CYT-2 (regulatory)-expressing cells. Together, these results indicate that Th1 switching to IL-10 expression is directly linked to the CBP, and that populations expressing IL-10 have higher levels of CBP-related genes when compared to IL-10-negative populations.

**Inhibition of the mevalonate pathway blocks Th1 switching**. To functionally assess the relationship between cholesterol biosynthesis and the generation of IL-10-expressing T cells, we blocked cholesterol biosynthesis during Th1 switching by treating cell cultures with atorvastatin, a synthetic lipid-lowering statin that competitively inhibits HMG-CoA reductase, one of the first steps of the mevalonate pathway (Supplementary Figure 2). Atorvastatin inhibited the generation of both IL-10-expressing double-positive (IFNγ+IL-10+) and single-positive (IFNγ−IL-10+) cells in a dose-dependent manner, while the frequency of IFNγ+IL-10− cells was increased (Fig. 2a, b and Supplementary Figure 3 for gating strategy), indicating that statin treatment blocks Th1 switching to IL-10. Inhibition of IL-10 levels in statin-treated cultures was prevented by supplementation with mevalonic acid (MA), confirming the specificity of this effect, and arguing against off-target effects of the pharmacological inhibitor. Measurement of IL-10 secretion after 36 h of culture produced similar results (Fig. 2c). It is noteworthy that atorvastatin had only a modest effect on the MFI of IL-10 in residual populations of IL-10-expressing cells (Supplementary Figure 4a), suggesting again that statins primarily block the transition from IFNγ to IL-10-producing cells, rather than

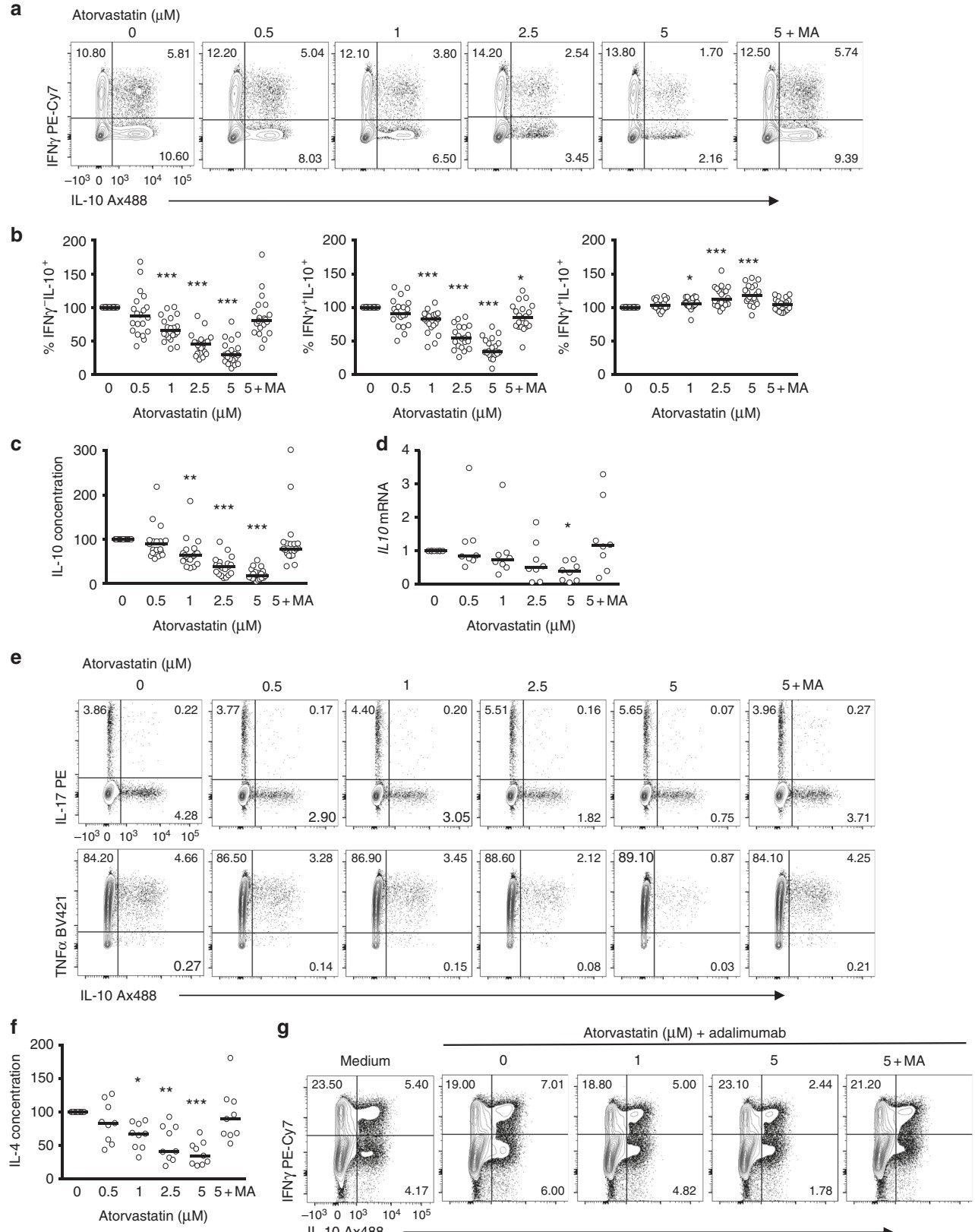

regulating levels of IL-10 within existing populations of IL-10-expressing T cells. Moreover, a reduction in *IL10* mRNA expression following statin treatment was detected (Fig. 2d), pointing to a link between CBP and the transcriptional regulation of IL-10 expression. In our experimental conditions, atorvastatin

did not induce changes in cell apoptosis or viability (Supplementary Figure 4b and 4c), nor could we detect cell proliferation at the 36 h time point, ruling out the possibility that reduced rates of cell proliferation by atorvastatin accounted for the changes in IL-10 expression (Supplementary Figure 4d). Likewise,

**Fig. 2** T-helper type 1 (Th1) switching to interleukin-10 (IL-10) is blocked when the mevalonate pathway is inhibited. Purified human CD4$^+$ T cells stimulated in vitro with plate-bound α-CD3 (2 μgml$^{-1}$) + α-CD46 (5 μgml$^{-1}$) and recombinant human interleukin-2 (rhIL-2) (50 Uml$^{-1}$) were cultured for 36 h in the presence of atorvastatin and 250 μM mevalonic acid (MA) as indicated, unless stated otherwise. **a** Representative flow cytometric analysis of intracellular interferon-γ (IFNγ) and IL-10 staining. **b** Normalised frequency of IFNγ$^-$IL-10$^+$- (left), IFNγ$^+$IL-10$^+$- (centre) and IFNγ$^+$IL-10$^-$- (right) producing cells ($n = 20$). **c** Normalised concentration of secreted IL-10 ($n = 19$). **d** Normalised *IL10* messenger RNA (mRNA) levels ($n = 8$). **e** Representative flow cytometric analysis of intracellular IL-17 (top) and tumour necrosis factor-α (TNFα) (bottom) co-stained with IL-10 of atorvastatin-treated cells (cumulative data shown in supplementary Fig. 3g). **f** Normalised concentration of secreted IL-4 ($n = 9$). **g** Effect of atorvastatin (AT) treatment on purified CD4$^+$ cells co-cultured with monocytes and α-CD3 (100 ng ml$^{-1}$) in the absence or presence of Adalimumab (1 μgml$^{-1}$). Data show a representative dot-plot for intracellular IL-10 and IFNγ production (cumulative data shown in Supplementary Fig. 3h). Graphs show independent donors (dots) normalised to 0 μM dose of atorvastatin; bars represent median values. *<0.05, **<0.01 and ***<0.001 denote a significant difference compared to untreated cells by repeated-measures one-way analysis of variance (ANOVA) test with post hoc Dunnett's correction (**b**, **d**, **f**) or Friedman test with post hoc Dunn's correction (**c**)

atorvastatin had minimal effects on CD4$^+$ T cell activation profiles based on the expression of the cell surface markers CD69 or CD25 (Supplementary Figure 4e and 4f).

Statins have previously been reported to alter Th effector cell responses[32,33], so we next determined the effect of atorvastatin on the expression profile of IL-17 and tumour necrosis factor-α (TNFα) in Th1 switching cells. While the results showed a decrease in the frequency of IL-17$^+$IL-10$^+$ and TNFα$^+$IL-10$^+$ cells with atorvastatin treatment, we documented a concomitant increase in IL-17 and TNFα single-positive populations that did not express IL-10 (Fig. 2e and Supplementary Figure 4g). Once again these effects were reversed by MA supplementation. During analysis of the effects of statin on Th2 cytokine profiles, we were unable to reliably detect intracellular IL-4 by flow cytometry. Instead, we measured secreted cytokine over the 36 h activation period. Like IL-10, levels of IL-4 were also decreased in the presence of atorvastatin (Fig. 3f), indicating that the effects of CBP perturbation are not confined to IL-10 expression, but may also influence Th2 responses, while at the same time sparing or even potentiating Th1 and Th17 effector cytokines.

To determine whether the effects of CBP perturbations on IL-10 were specific for CD46-dependent induction, we examined the effects of statin on α-CD3-stimulated CD4$^+$ T cells co-cultured with monocytes in the presence of adalimumab, a humanised α-TNFα monoclonal antibody (mAb), as an alternative pathway to IL-10 induction[34,35]. In keeping with the results above, we documented dose-dependent inhibition of IL-10-producing T cells, while the frequency of IFNγ$^+$ cells remained unchanged (Fig. 2g and Supplementary Figure 4h). Similar results were obtained in an IFNα-dependent IL-10-inducing system[12] when cells were cultured in the presence of atorvastatin ± MA (Supplementary Figure 4i). Together, these data further support the notion that the CBP regulates IL-10 expression in human CD4$^+$ T cells.

**IL-10 is not dependent on isoprenylation or vitamin D3.** We next set out to define more precisely which elements of the CBP regulate IL-10. First, we ascertained that statins exerted their inhibitory mechanism by targeting intracellular pathways through a comparison of the effects of atorvastatin, mevastatin and pravastatin. Similarly to atorvastatin, mevastatin is a lipophilic statin preparation that diffuses across the membrane, while pravastatin is lipophobic and cannot enter the cell in the absence of active transport[36,37]. Similar inhibitory effects were observed on T cells stimulated in the presence of mevastatin but not pravastatin (Fig. 3a), demonstrating that statins mediate their IL-10 inhibitory effects through mechanisms that require targeting of intracellular pathways.

To unravel the specific branch of the CBP that regulates IL-10 expression, we cultured Th1 switching cells in the presence of pharmacological inhibitors that target enzymes in the cholesterol

pathway (Supplementary Figure 2) and analysed IL-10 expression. Firstly, we tested whether atorvastatin inhibits IL-10 by inhibiting isoprenylation. Isoprenylation regulates cell signalling cascades, and targeted inhibition of the isoprenylation pathway is known to alter cytokine expression patterns in vivo and in vitro[32]. To this end, we incubated Th1 switching cells with inhibitors of farnesyl-PP transferase (FTase I inhibitor), geranylgenaryltransferase I (GGTI-298) and Rab geranylgeranyl transferase (psoromic acid) and assessed IL-10 expression. None of these compounds exerted statin-like inhibitory effects on either the frequency of IL-10$^+$ cells (Fig. 3b) or its secretion (Supplementary Figure 5a). In keeping with these findings, supplementation of atorvastatin-treated cells with increasing concentrations of the metabolites farnesyl-PP or geranylgeranyl-PP failed to rescue the frequency of IL-10$^+$ cells to those levels observed with MA supplementation (Fig. 3c), a finding also confirmed by enzyme-linked immunosorbent assay (ELISA) (Supplementary Figure 5b). Thus, the inhibitory effects of short-term atorvastatin exposure do not appear to be related to perturbations of the isoprenylation pathway.

The mevalonate pathway generates 7-dehydrocholesterol, the precursor of vitamin D3, a metabolite that has been described as key in the regulation of IL-10 in Th cells[28]. We therefore tested whether inhibition of 7-dehydrocholesterol biosynthesis by the small-molecule inhibitor U18666A would prevent Th1 switching to IL-10. We observed a consistent reduction in IL-10 expression, most notable at the highest concentration of inhibitor tested (Fig. 3d and Supplementary Figure 5c). To address the specificity of this effect, we supplemented T cell cultures with increasing concentrations of calcitriol, the active metabolite of vitamin D3, in the presence of 5 μM atorvastatin. At concentrations known to induce IL-10 production in human CD4$^+$ T cells[28], calcitriol failed to restore IL-10 levels to those achieved with MA at any concentration (Fig. 3e and Supplementary Figure 5d), indicating that deficiency of vitamin D3 is not responsible for the observed regulation of IL-10 by statin treatment, and implicating other U18666A targets, such as NCP1[38], in the regulation of IL-10.

We also ruled out the possibility that atorvastatin treatment substantially altered the cellular lipid expression profile. We performed lipidomics analyses in Th1 switching cells treated with different doses of atorvastatin with or without MA by liquid chromatography mass spectrometry. We analysed the content of 536 lipid species covering the main lipid families (triacylglycerides, sphingomyelins, phosphatidylcholines, phosphatidylserines, ceramides) (Supplementary Data 2). PCAs of the dataset revealed that samples clustered by donor rather than treatment (Supplementary Figure 5e), indicating that in our system statins are not substantially perturbing the levels of these lipid species within the timeframe of the experiments. Of note, cholesterol levels were not changed in the presence of atorvastatin (Fig. 3f). These data suggested that perturbations of the whole metabolic pathway,

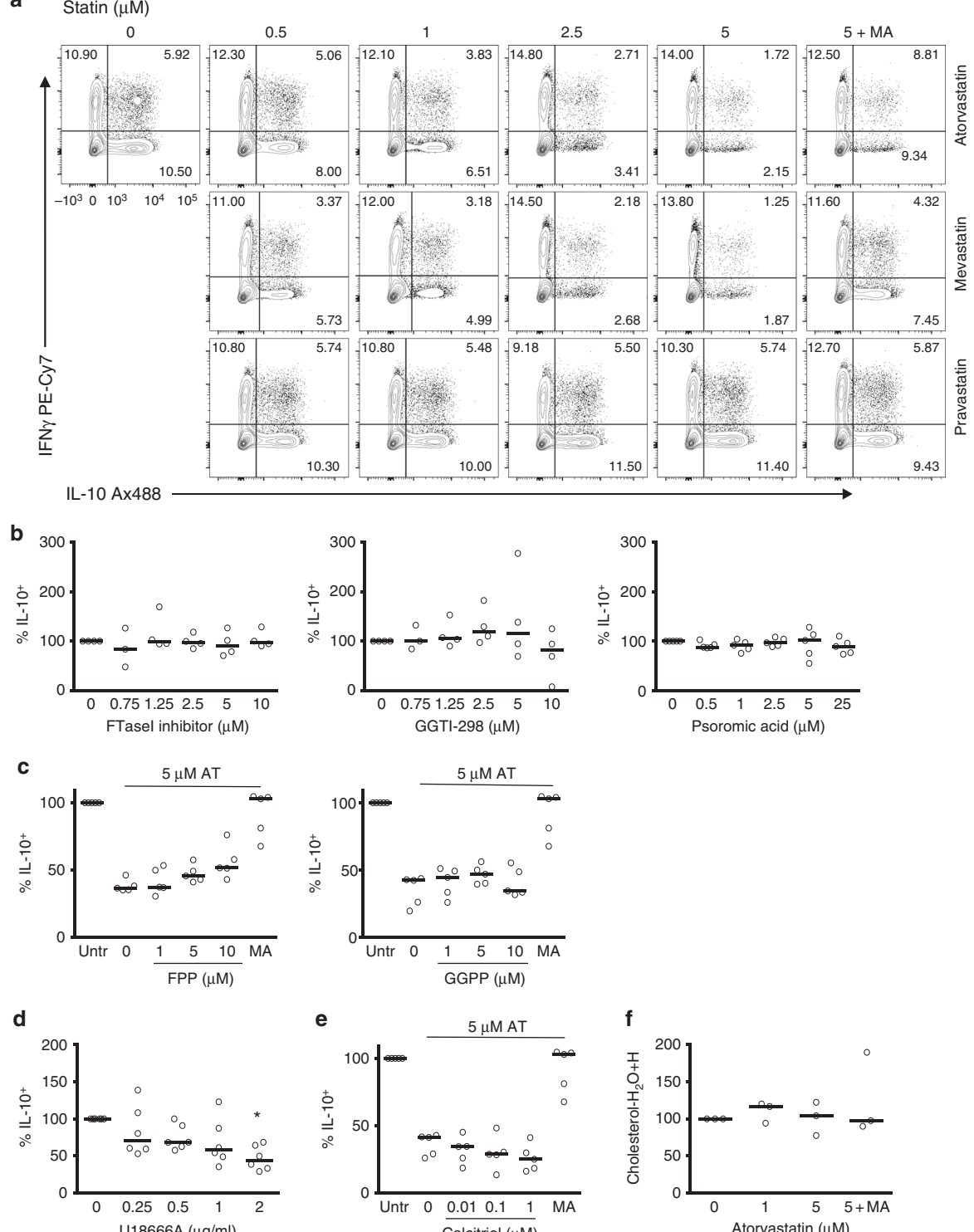

rather than a specific metabolite, might be responsible for the reduction in IL-10 in Th1 switching cells.

**25-HC alters cholesterol homeostasis**. A failure to regulate IL-10 expression via manipulation of pathway branch points described above, such as those linked to isoprenylation or vitamin D biosynthesis, provided the first clues that intact cholesterol flux may be a key requirement for robust Th1 switching[39]. To test whether coordinate reduction in levels of CBP enzymes would be sufficient to uncouple IL-10 production from the Th1 switching

programme, we studied the effects of 25-HC supplementation in our culture system. 25-HC is an immune oxysterol derived from cholesterol that is abundantly secreted by activated macrophages and is known to control cholesterol flux by inhibiting 3-hydroxy-3-methylglutaryl-CoA reductase and the SREBP-2 pathway, responsible for the transcriptional regulation of the CBP machinery (Supplementary Figure 2)[29,30]. First, we addressed if increasing concentrations of 25-HC within the physiological range would exert a negative effect on CBP homeostasis during Th1 switching by measuring key CBP enzymes known to be

**Fig. 3** Interleukin (IL-10) regulation in T-helper type 1 (Th1) switching cells is not dependent on isoprenylation, vitamin D3 or cellular cholesterol content. Purified human CD4$^+$ T cells stimulated in vitro with plate-bound α-CD3 (2 μgml$^{-1}$) + α-CD46 (5 μgml$^{-1}$) and recombinant human interleukin-2 (rhIL-2) (50Uml$^{-1}$) were cultured for 36 h in the presence of selected metabolites and inhibitors as indicated. **a** Representative flow cytometric analysis of intracellular interferon-γ (IFNγ) and IL-10 staining of cells treated with different formulations of statin (data representative of three independent donors). **b** Normalised frequency of IL-10$^+$ cells cultured in the presence of inhibitors for farnesyl-PP transferase (FTase I inhibitor) (left) (n = 4), geranylgeranyl transferase I (GGTI-298) (centre) (n = 4) and Rab geranylgeranyl transferase (psoromic acid) (right) (n = 5). **c** Normalised frequency of IL-10$^+$ cells cultured with 5μM atorvastatin (AT) and increasing concentrations of farnesylpyrophosphate (FPP) (left) and geranylgeranylphyrophosphate (GGPP) (right) and mevalonic acid (MA) as control (n = 5). **d** Normalised frequency of IL-10$^+$ cells cultured in the presence of increasing concentrations of U18666A (n = 6). **e** Normalised frequency of IL-10$^+$ cells cultured with 5 μM atorvastatin (AT) and increasing concentrations of calcitriol and mevalonic acid (MA) as a control (n = 5). **f** Normalised total cellular cholesterol content measured by liquid chromatography mass spectrometry. The [cholesterol −H$_2$O + H]$^+$ ion was the dominant adduct for cholesterol and relative intensity is shown for this ion (n = 3). Graphs show independent donors (dots) normalised to 0μM dose of atorvastatin; bars represent median values. *<0.05 denote a significant difference compared to untreated cells by Friedman test with post hoc Dunn's correction (**d**)

---

regulated by SREBP-2 and which we had also determined to be related to IL-10 switching (Fig. 1b and Supplementary Figure 2). Expression of *LDLR*, *HMGCS1*, *FDFT1* and *DHCR7* were downregulated at the mRNA level in a dose-dependent manner when cells were exposed to 25-HC for the 36 h incubation period (Fig. 4a). These data demonstrate that 25-HC perturbs CBP homeostasis in CD4$^+$ T cells. Unexpectedly, supplementation of 25-HC-treated cultures with cholesterol restored the levels of SREBP-2 target genes (Supplementary Figure 6a). This observation suggests that cholesterol interferes with 25-HC regulation of SREBP-2 transcriptional activity in our system.

We next investigated how 25-HC regulates IL-10 expression. In agreement with our hypothesis, supplementation of Th1 switching cells with increasing concentrations of 25-HC led to a dose-dependent inhibition of IL-10 expression as determined by flow cytometry (Fig. 4b, c) and by quantification of secreted protein levels (Fig. 4d), while the levels of IFNγ remained unchanged (Fig. 4b, c). The specific suppression of IL-10 expression by 25-HC was also observed in other IL-10-inducing in vitro systems (Supplementary Figure 6c and 6d)[12,34].

To further verify the requirement of cholesterol flux in IL-10 expression, we performed experiments in which cells were cultured either under sub-optimal lipid conditions (media supplemented with delipidised serum, or lipid-free media) or where cultures were supplemented with cholesterol. Culturing cells in lipid-free media had a significant impact on downregulating IL-10 expression, while IFNγ levels remained unchanged (Fig. 4e). Supplementation with cholesterol had no additive effect on IL-10 expression in cells cultured in standard media conditions. However, when cholesterol was added back to cultures in the presence of high 25-HC concentrations, Th1 switching to IL-10 was fully restored (Fig. 4e), while supplementation with MA did not prevent the inhibitory effects of 25-HC on IL-10 expression levels (data not shown). Taken together, these data demonstrate that inhibition of Th1 switching by 25-HC is linked to inhibition of cholesterol flux through similar mechanisms to those described in innate immune cells[30] and that optimal cholesterol homeostasis is a requirement for adequate IL-10 expression by human CD4$^+$ T cells.

**Cholesterol metabolism regulates c-Maf expression.** Because our data suggested that CBP regulates IL-10, at least in part, at the transcriptional level (Fig. 2d), we set out to identify the transcription factor/s that might be responsible for the CBP-mediated IL-10 downregulation, reasoning that CBP flux regulates factors directly involved in the transcriptional regulation of IL-10. To test this hypothesis, we cultured Th1 switching cells in the presence of increasing concentrations of 25-HC and measured *MAF*, the gene that encodes for the transcription factor c-Maf, considered as the master transcription regulator for IL-10 in

CD4$^+$ T cells[10,15,17,40]. Blockade of the CBP with 25-HC reduced *MAF* mRNA levels in a dose-dependent manner (Fig. 5a). However, mRNA levels of *PRDM1*, which encodes the transcription factor Blimp-1 and which has recently been implicated in sterol-mediated regulation of IL-10 in mouse CD4$^+$ T cells[41], did not change with 25-HC treatment (Fig. 5b). Downregulation of *MAF* at the mRNA level was also evident at the protein level (Fig. 5c, d) and this was prevented upon supplementation of cultures with cholesterol (Fig. 5e). Moreover, there was a strong correlation between the protein levels of IL-10 and c-Maf (Fig. 5f), suggesting that these events are related.

**Cholesterol flux predicts progression to inflammatory disease.** The data above demonstrated that alterations in the CBP regulate pathways linked to immune resolution, at least in vitro, raising the possibility that perturbations in cholesterol flux could promote the inflammatory response in human disease. To explore this in an in vivo context, we tested the hypothesis that expression of CBP genes in synovial tissue biopsies from subjects at high risk of developing RA could predict disease outcome. Specifically, we reasoned that high levels of cholesterol-25-hydroxylase (*CH25H*), the enzyme that synthesises 25-HC from cholesterol, would be associated with failure to resolve inflammatory responses and progression to clinically apparent inflammatory arthritis. To this end, we interrogated gene expression profiling data from an exploratory prospective study in which synovial tissue biopsies we obtained from 13 autoantibody-positive arthralgia patients deemed to be "at risk" of developing RA[42,43]. Over a median follow-up time of 20 months (interquartile range (IQR) 2–44) six individuals developed RA, while seven individuals did not develop RA over a median follow-up time of 85 months (IQR 69–86). The wide IQR for developing disease reflects the variable progression rates (i.e. time to develop arthritis from baseline). We stratified the RA-risk group into low and high expressers according to synovial *CH25H* mRNA expression. All high-risk individuals who progressed to inflammatory arthritis displayed high mRNA levels of *CH25H*, while the *CH25H*$^{lo}$ group remained arthritis free for the duration of follow-up (Fig. 6a, left; hazard ratio (HR) 5.3, 95% confidence interval (CI) 1.0, 27.9; P = 0.047). Further analysis revealed that disease outcomes could also be stratified according to synovial expression of *DHCR7* (Fig. 6a, centre, HR 8.9, 95% CI 1.6, 48.7; P = 0.012) and *FDFT1* (Fig. 6a, right, HR 14.5, 95% CI 2.7, 78.7; P = 0.002). In contrast to *CH25H*, low expression of these CBP enzymes was found to be associated with low IL-10 expression during Th1 switching, and was found to be associated with arthritis development. Despite the limitations of this small sized-study, these data support a model in which perturbations of cholesterol flux indicators are linked to disease progression during the earliest detectable phase of pathological inflammatory responses in vivo.

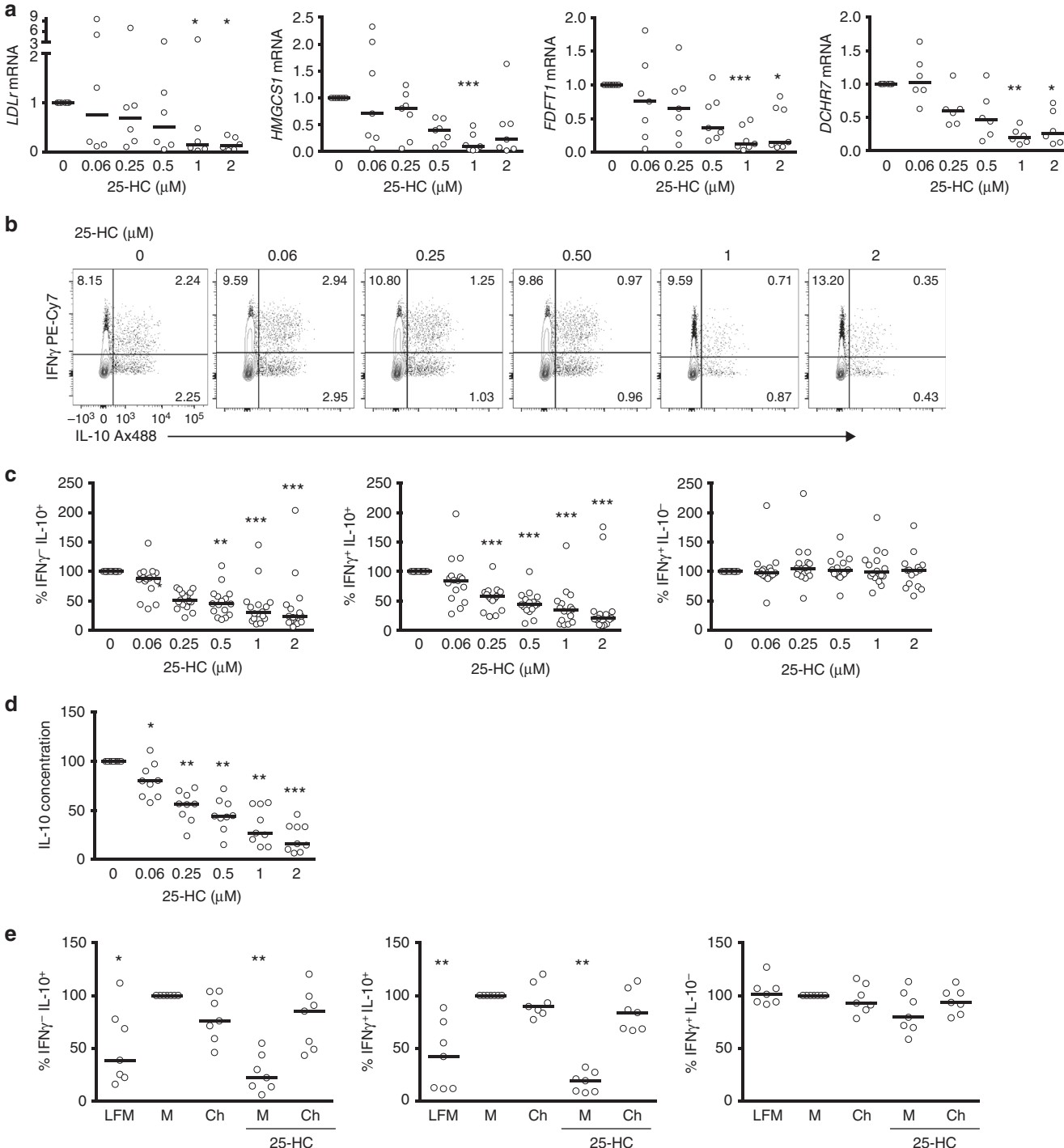

**Fig. 4** Interleukin (IL-10) expression in T-helper type 1 (Th1) switching cells is dependent on intact cholesterol pathway fitness. Purified human CD4+ T cells stimulated in vitro with plate-bound α-CD3 (2 μgml$^{-1}$) + α-CD46 (5 μgml$^{-1}$) and recombinant human interleukin-2 (rhIL-2) (50 Uml$^{-1}$) were cultured for 36 h in the presence of 25-hydroxycholesterol (25-HC). **a** Normalised *LDLr*, *HMGCS1*, *FDFT1* and *DCHR7* messenger RNA (mRNA) levels (*n* = 6–7). **b** Representative flow cytometric analysis of intracellular interferon-γ (IFNγ) and IL-10 staining. **c** Normalised frequency of IFNγ⁻IL-10⁺- (left), IFNγ⁺IL-10⁺- (centre) and IFNγ⁺IL-10⁻- (right) producing cells (*n* = 16). **d** Normalised concentrations of secreted IL-10 (*n* = 9). **e** Normalised frequency of IFNγ⁻IL-10⁺- (left), IFNγ⁺IL-10⁺- (centre) and IFNγ⁺IL-10⁻- (right) producing cells cultured under lipid-free medium (LFM), fully supplemented medium (M) and cholesterol (500×) (Ch) in the presence or absence of 25-HC (2μM) for 36 h (*n* = 7). Graphs show independent donors (dots) normalised to untreated cells; bars represent median values. *<0.05, **<0.01 and ***<0.001 denote a significant difference compared to untreated cells by Friedman test with post hoc Dunn's correction (**a**, **c**: IFNγ⁻IL-10⁺, **e**) or by repeated-measures one-way analysis of variance (ANOVA) test with post hoc Dunnett's correction (**c**: IFNγ⁺IL-10⁺ and IFNγ⁺IL-10⁻, **d**)

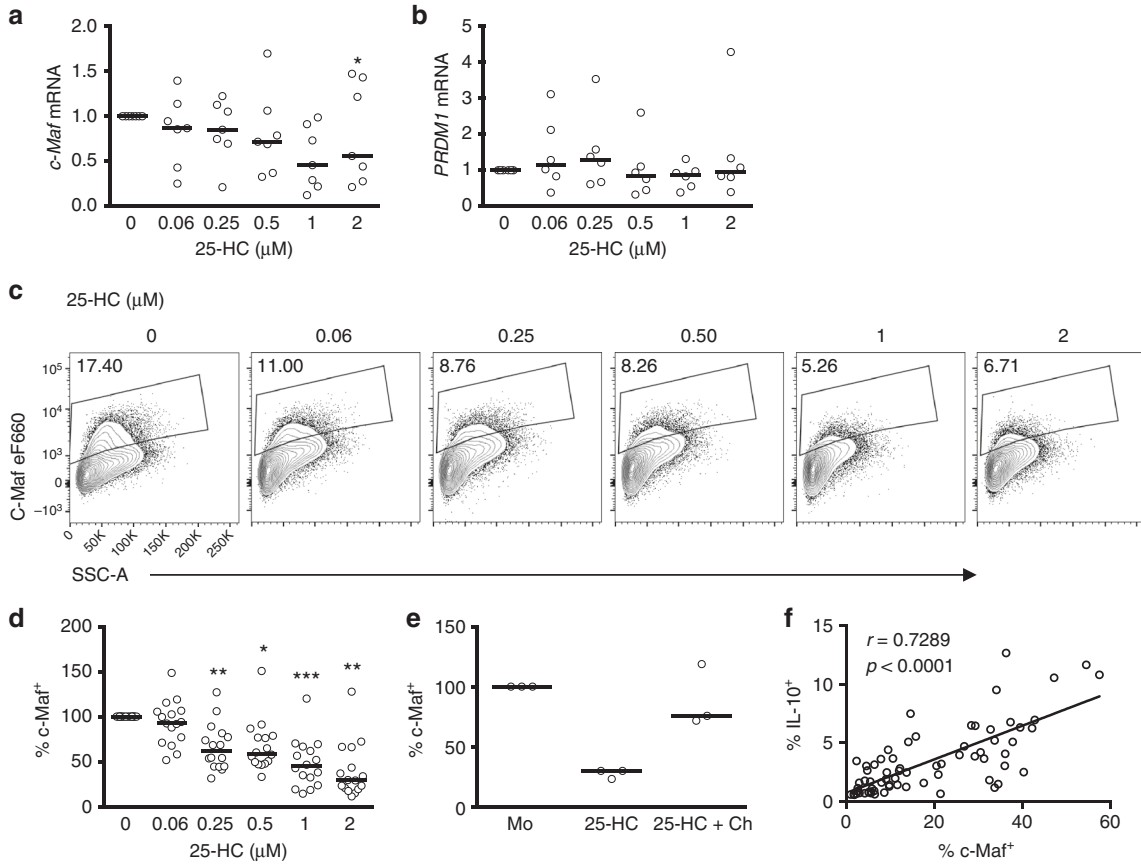

**Fig. 5** Cholesterol biosynthesis pathway inhibition interferes with c-Maf expression. Purified human CD4$^+$ T cells stimulated in vitro with plate-bound α-CD3 (2 µgml$^{-1}$) + α-CD46 (5 µgml$^{-1}$) and recombinant human interleukin-2 (rhIL-2) (50 Uml$^{-1}$) were cultured for 36 h in the presence of 25-hydroxycholesterol (25-HC). **a** Expression levels of *MAF* mRNA ($n = 7$). **b** Expression levels of *PRDM1* mRNA ($n = 6$). **c** Representative flow cytometric analysis of intracellular c-Maf staining. **d** Normalised frequency of c-Maf$^+$ cells ($n = 16$). **e** Normalised frequency of c-Maf$^+$ cells cultured under fully supplemented medium (M) and cholesterol (500×) (Ch) in the presence or absence of 2µM 25-HC ($n = 3$). **f** Correlation between frequency of c-Maf$^+$ and IL-10$^+$ cells, numbers denote $r$ and $p$ values for Spearman's correlation test. Graphs show independent donors (dots) normalised to untreated cells; bars represent median values. *<0.05 and **<0.01 denote a significant difference compared to untreated cells by Friedman test with post hoc Dunn's correction (**a**) or by repeated-measures one-way analysis of variance (ANOVA) test with post hoc Dunnett's correction (**d**)

## Discussion

Tight control of immune effector responses promotes homeostasis and health. Active acquisition of a resolution phase by effector T cells, through induction of IL-10 expression, is critical to this process[2]. Here, we undertook a systematic and unbiased approach to uncover novel pathways that control resolution of immune responses within the human CD4$^+$ T cell compartment and discovered that IL-10 is metabolically regulated by the CBP in human CD4$^+$ T cells. Our experiments have shown that perturbation of cholesterol biosynthesis by statins or 25-HC leads to highly specific inhibition of IL-10 expression, an effect that was reversed upon metabolite supplementation. Moreover, our data uncovered a role for the CBP in controlling c-Maf expression, since uncoupling of the pathway decreased expression of this transcription factor, a master regulator of IL-10 expression in immune cells[40].

Unsupervised analysis revealed that the CBP is intimately linked to IL-10 expression in primary human CD4$^+$ T cells. In keeping with these findings, blockade of the pathway with statins in in vitro culture showed a robust inhibition of IL-10 expression, pointing to a novel and unexpected anti-resolution effect of the drug during Th1 switching. The effects of statins on immune and inflammatory responses are conflicting. On the one hand, pharmacological inhibition of cholesterol biosynthesis with statins has been reported to have predominantly anti-inflammatory

effects[44–46]. In keeping with this, murine effector CD4$^+$ T cells treated with statins secrete less IFNγ and more IL-4[47,48] by reducing biosynthesis of isoprenoids[32]. On the other hand, a more pro-inflammatory role has been reported, where statins increase IFNγ and TNFα secretion[33,49], independent of cell cholesterol content[33], while other studies have reported reduced IL-10 expression in both human and murine T cells upon exposure to statins[50–52]. In our experiments, short-term exposure (36 h) to atorvastatin was sufficient to uncouple IFNγ and IL-10 production during Th1 switching without discernible effects on cell proliferation, cell viability, isoprenylation or cellular lipid content. This leads us to conclude that blocking the transition to IL-10 expression is one of the earliest consequences of a change in flux along the CBP, and to speculate that IL-10 may not be regulated by the actions of a specific metabolite, but linked to states of metabolic flux through the cholesterol pathway.

Further evidence to support the altered cholesterol metabolic flux model is the observation that immune oxysterols also inhibit Th1 switching, since it is known that 25-HC reduces cholesterol flux in T cells by blocking the nuclear translocation of SREBP-2[30], hence blocking the transcription of key components of the CBP, an effect that we have also reported here. The production of 25-HC is tightly regulated, and produced in the endoplasmic reticulum upon induction of CH25H[29]. The enzyme is strongly upregulated by type I IFN in myeloid cells[53] and plays a key role

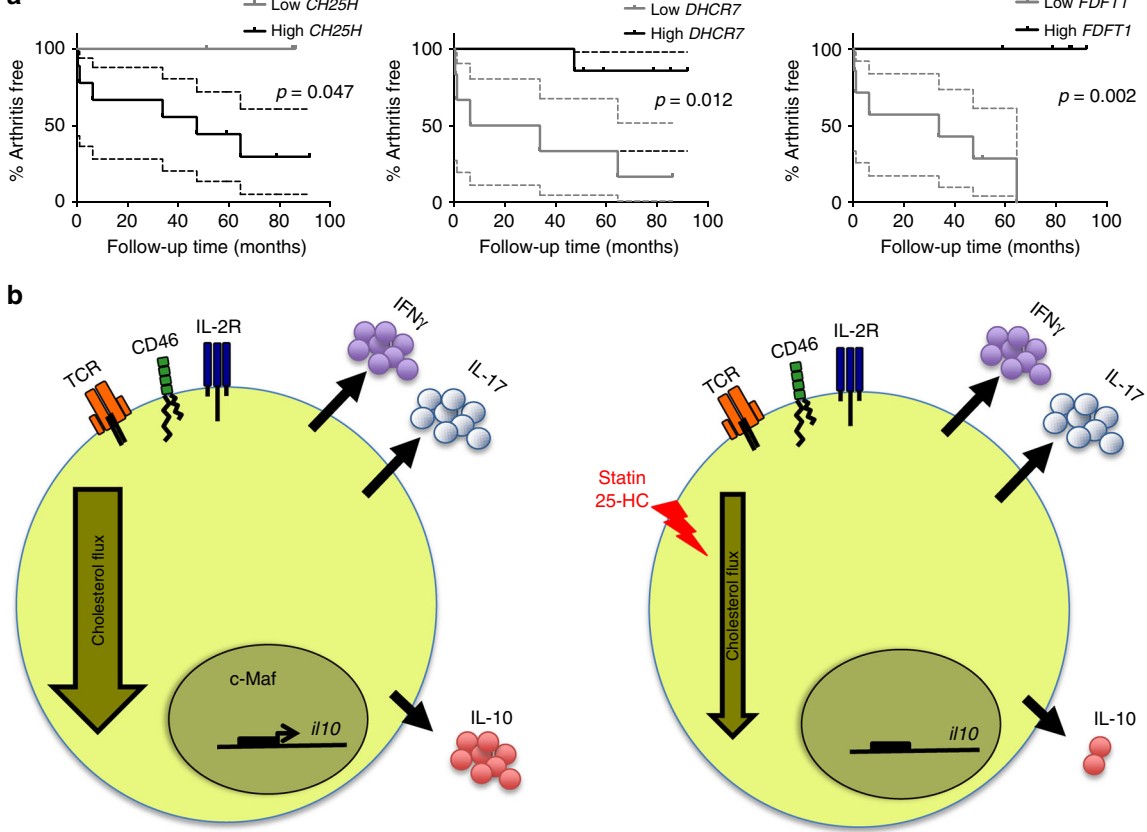

**Fig. 6** Messenger RNA (mRNA) levels of cholesterol biosynthesis pathway (CBP) enzymes are related to disease progression in man. **a** Expression levels of cholesterol-25-hydroxylase (CH25H), FDFT1 and DHCR7 obtained from a gene expression profiling study of synovial biopsies from 13 rheumatoid arthritis (RA)-risk individuals who were followed over time to investigate the development of arthritis. RA-risk individuals were stratified into relative low or high expressers for the indicated genes, after which their arthritis-free survival was compared using a log-rank (Mantel–Cox) test. Graphs display arthritis-free survival curves including the 95% confidence interval (CI). **b** Schematic representation of interleukin-10 (IL-10) regulation by the cholesterol biosynthesis pathway. Normal cholesterol pathway activity is required for adequate transcription of the IL10 gene and IL-10 expression in T-helper effector cells (left); when the flux is reduced via statin or 25-hydroxycholesterol (25-HC) supplementation, c-Maf expression is downregulated and IL-10 gene transcription reduced

in the anti-viral response. Interestingly, the production of 25-HC and the expression of CH25H have been reported recently in adaptive immune cells[41]. Strikingly, 25-HC produced by mouse $CD4^+$ T cells inhibited IL-10 expression in these cells by reducing the expression of the transcription factor Blimp-1[41], a change that we did not detect in human $CD4^+$ T cells. This report confirms our observation that oxysterols regulate the expression of IL-10 in $CD4^+$ T cells, although via a different transcriptional mechanism, providing evidence of differences in the IL-10 regulatory mechanisms between mouse and human.

As CBP perturbation significantly reduced IL10 mRNA levels, we hypothesised that transcription factors reported to regulate IL-10 expression were also metabolically regulated. c-Maf has been described as the master transcription factor that drives Th2 cytokines, including IL-10, in $CD4^+$ T cells[13,17] and more recently in Tregs[54]. Our results show that perturbations in CBP affect c-Maf levels in the same way as IL-10, suggesting that c-Maf might acts as a molecular platform that links metabolism and immune resolution. In keeping with these findings was the observation that statin treatment blocks IL-4 secretion. Defining the molecular link between c-Maf and cholesterol metabolism requires more investigation, but consistent with our observations, small members of the Maf family have been shown to form heterodimers with the active form of the transcription factor nuclear factor erythroid-derived 2-related factor 1[55] recently described as a cholesterol sensor in the endoplasmic reticulum[56].

Moreover, chromatin immunoprecipitation deep sequencing approaches to identify SREBP-2 binding sites have revealed Maf family members as potential targets[57], providing a potential molecular link between Maf family members and the CBP.

We believe that our findings shed light on disease mechanisms that may extend beyond regulation of Th1 switching at the cellular level. Firstly, IFN-inducible gene signatures have been identified during the early phase of immune-mediated inflammatory diseases[61,62], and so it is conceivable that IFNα-induced CH25H/25-HC plays a key role in the initiation and persistence of inflammatory responses by uncoupling IL-10-dependent resolution. Secondly, the links between statin use and infection or cancer have been attributed to their lipid-lowering effects or to attenuation of the prenylation branch, leading to perturbations in cell proliferation and signalling. Here, we offer an alternative but not mutually exclusive mechanism for enhanced host and anti-tumour immunity, underpinned by the capacity of T cells to generate more IFNγ relative to IL-10. Finally, clinical trials of biological therapy in patients with RA have demonstrated reduced serum cholesterol levels associated with active inflammatory disease, and increases in lipids in patients responding to TNF inhibitors, as the inflammation resolves. This observation, the so-called "lipid paradox", has been linked to chronic inflammatory states[58], and initially raised concerns about the risks of cardiovascular comorbidity with long-term biological therapy. It is conceivable that rather than increasing

cardiovascular risk, increased serum cholesterol and cellular cholesterol flux contribute to mechanisms whereby TNF blockade increases IL-10 production by T cells[34,59]. Conversely, rare monogenic diseases that result in substantial reduction of cholesterol biosynthesis, such as mevalonate kinase deficiency, are characteristically inflammatory[60,61], providing another link between cholesterol metabolism and inflammatory responses.

The mechanisms whereby perturbation of cholesterol biosynthesis blocks IL-10 and c-Maf expression remain to be determined. It will also be important to discern whether the mechanism is common to all IL-10-expressing cell subsets, or is unique to CD4+ T cells. This knowledge can then be used as a framework to explore in relevant models the immunomodulatory properties of small-molecule inhibitors and metabolites that regulate cholesterol flux. Experience to date demonstrates that the CBP is drug-able. This opens up new therapeutic avenues to re-purpose existing drugs, as well as develop new ones, for fine-tuning the sterol pathway according to whether the therapeutic goal is to prevent chronic inflammation, or to promote host or tumour immunity.

## Methods

**Cell purification and culture**. CD4+ T cells were positively isolated from density gradient centrifugation (Lymphoprep, Axis-Shield) preparations of peripheral blood mononuclear cells obtained from healthy volunteers (approval REC06/Q0705/20 obtained from Bromley Research Ethics Committee) and NC24 CD Leucocyte Cones (NHS Blood and Transplant) using CD4 microbeads (Miltenyi Biotec) following the manufacturer's recommendations. Purity of CD4+ cells was routinely tested as >98%. For inducing Th1 switching cells, CD4+ T cells were cultured at $1.5 \times 10^6$ ml$^{-1}$ in 48-well plates ($3 \times 10^5$ cells per well) coated with mAbs to CD3 (Clone OKT3, 2 µgml$^{-1}$) and CD46 (clone TRA-2-10, 5 µgml$^{-1}$) both from BioLegend[18]. Cells were cultured in RPMI-1640 containing 2 mM L-glutamine and 0.1 g l$^{-1}$ sodium bicarbonate (Sigma) supplemented with 10 mM HEPES (PAA), 100 Uml$^{-1}$ penicillin + 0.1 mg ml$^{-1}$ streptomycin (PAA) and 10% foetal bovine serum (Sigma) (fully supplemented media) or 10% lipid depleted foetal bovine serum (Biowest) (lipid-free media), in the presence of 50 Uml$^{-1}$ of recombinant human IL-2 (Proleukin, Novartis). Where indicated, cells were cultured in the presence of atorvastatin (Calbiochem), (R)-MA lithium salt (Sigma), mevastatin (Sigma) (kindly donated by Dr. P. Vantourout), pravastatin (Sigma), FTase I inhibitor (Calbiochem), GGTI-298 (Calbiochem), psoromic acid (Santa Cruz Biotechnology), farnesylpyrophosphate ammonium salt (Sigma), geranylgeranyl pyrophosphate ammonium salt (Sigma), U18666A (Sigma), 1α,25-dihydroxyvitamin D$_3$ (calcitriol) (Sigma), 25-hydroxycholesterol (Avanti lipids) and/or "soluble" cholesterol (SyntheChol® NS0 Supplement, Sigma). Carrier/diluent controls were performed and no differences in cell viability were detected (data not shown)[62]. Doses of cholesterol pathway inhibitors and metabolites have been previously reported as active in in vitro experiments using human cells[62–64]. For isolation of cytokine-producing populations, activated cells were stained with both IL-10 and IFNγ Cytokine Secretion Assay kit (Miltenyi Biotec) following the manufacturer's protocol. Cytokine-positive cells were consequently FACS sorted using a FACS Aria (BD Biosciences). For adalimumab experiments, CD4+ T cells and CD14+ monocytes ($1 \times 10^6$ ml$^{-1}$) were co-cultured with 100 ng ml$^{-1}$ CD3 mAb and 1 µgml$^{-1}$ adalimumab (Abbott) for 3 days[34,35]. For IFNα-inducing experiments, CD4+ T cells ($2.5 \times 10^6$ ml$^{-1}$) were cultured with plate-bound CD3 (0.5 µgml$^{-1}$), soluble CD28 (clone CD28.2, BD Biosciences, 1 µgml$^{-1}$) and IFNα 2a (PBL Assay Science, 600 Uml$^{-1}$) for 3 days[12]. Informed, written consent was obtained from all donors.

**Microarray**. For primary cell experiments, cell pellets for mRNA isolation were lysed in Trizol (Ambion, Life Technologies) and RNA extracted by guanidium-isothiocyanate phenol/chloroform extraction method following the manufacturer's instructions. Total RNA was quantified using the Nanodrop ND1000 and RNA integrity assessed with the Agilent Bioanalyzer 2100 using the RNA 6000 Nano Chips. One hundred nanograms of total RNA was used to prepare the targets, using the 3′ IVT Express kit (Affymetrix) in accordance with the manufacturer's instructions. Hybridisation cocktails were hybridised onto the Human Genome U133 plus2 gene chip (Affymetrix). For Jurkat cell line experiments, three technical replicates of Jurkat T cell stably expressing CYT-1 or CYT-2 and untransduced cells as control were analysed. RNA was converted to cDNA using the Ovation® Pico WTA Systems V2 and labelled using the Encore® BiotinIL Module (NuGen) according to the manufacturers recommendations. Labelled cDNA was hybridised onto Illumina HT12v4 arrays (Illumina).

**Flow cytometry assays and intracellular staining**. After the indicated culture time, cells were harvested. Surface staining for activation markers was performed

with CD69 (clone FN50) and CD25 (clone M-A251), both from BioLegend, after staining with Fixable Viability Dye eFluor780 (eBiosciences) to exclude dead cells. For intracellular staining of cytokines, cells were restimulated for 3 h with phorbol myristate acetate (50 ng ml$^{-1}$) and ionomycin (1µM) (Sigma) in the presence of Brefeldin A and GolgiStop (1000×, BD Biosciences). After stimulation, cells were stained with Fixable Viability Dye eFluor780 (eBiosciences), fixed in 4% paraformaldehyde solution (Electron Microscopy Sciences) and permeabilised in 0.1% saponin (Sigma) solution for intracellular staining with the following antibodies: IL-10 (clone JES3-9D7); IFNγ (clone 4S.B3); TNFα (clone Mab11) and IL-17 (clone BL168), all from BioLegend. For intracellular staining of c-Maf (clone sym0F1 from eBiosciences), the transcription factor staining buffer set from eBiosciences was used, following the manufacturer's instructions. FACS files were acquired on a FACS Canto (BD Biosciences) and analysed using the FlowJo software (TreeStar). Doublets and cells positive for Fixable Viability Dye eFluor780 were excluded from analyses. Apoptosis was assessed by Annexin V staining (BD Biosciences) on treated cells following the manufacturer's protocol. Cell proliferation measurements were performed on cell trace violet (Invitrogen) stained cells following the manufacturer's protocol, after 36 h of incubation with plate-bound CD3 and CD46 or beads coated with CD3 and CD28 (Gibco) at 1:10 ratio in the presence of 50Uml$^{-1}$ of IL-2.

**Detection of soluble cytokines**. Cell culture supernatants were collected and stored at −20 °C until analysed. Cytokine levels were measured using specific ELISA kits for human IFNγ and IL-10 (R&D Systems) following the manufacturer's specifications. Colorimetric determinations were transformed into concentrations via interpolation of standard curves using a four-parameter logistic curve fit. For IL-4 measurements, a BD Cytometric Bead Array kit (BD Biosciences) was used and data were analysed according to the manufacturer's protocol.

**Quantitative real-time PCR**. Cell pellets for mRNA isolation were lysed in Trizol (Ambion, Life Technologies) and RNA was extracted by guanidium-isothescyanate phenol/chloroform extraction method following the manufacturer's protocols. Total RNA was quantified using the Nanodrop ND1000. Normalised RNA quantities were reverse transcribed into cDNA using the qPCRBIO Synthesis kit (PCR Biosystems). qPCR was performed in qPCRBIO Probe Mix (PCR Biosystems) using pre-validated FAM-labelled Taqman Gene Expression Assays for LDLr (Hs00181192_m1), HMGCS1 (Hs00940429_m1), FDFT1 (Hs00926054_m1) and DHCR7 (Hs01023087_m1), IL10 (Hs00961622_m1), MAF (Hs04185012_s1) and PRDM1 (Hs00153357_m1). The reactions were multiplexed using VIC-labelled 18S probe (Applied Biosystems), which was used for normalisation of Ct values. The reactions were run using the ABI Prism 7700 Sequence Detection System (Applied Biosystems). Ct values were determined using the SDS software (Applied Biosystems) and gene expression levels were determined according to the dCt method (relative abundance = $2^{(-\mathrm{dct})}$).

**Mass spectrometry assays**. Cell pellets were prepared from Th1 switching cells with the appropriate treatment. Lipids were extracted by liquid–liquid extraction in chloroform:methanol:water (2:1:1, v/v) using Folch's method[65]. The organic layer was removed, dried down under nitrogen and reconstituted in 15 mM ammonium acetate in isopropanol:methanol (2:1). An internal standard mix containing isotopically labelled lipids across several lipid classes was added. Samples were infused using a Triversa Nanomate® (Advion BioSciences, Ithaca, NY, USA), with capillary voltage 1.4 kV, 0.3 gas flow for 1 min into an LTQ Orbitrap Elite™ (Thermo Fisher Scientific). Spectra were acquired in positive and negative mode, from 200 to 1000 m/z. Data were converted to mzML format for subsequent data processing, and features were picked using an in-house R script. Lipid identity was performed by accurate mass using the LipidMaps database[66]. Basal levels of lipids for each donor were subtracted prior to multivariate statistical analysis. PCA was performed using SIMCA 14 (Umetrics, Sweden), following normalisation to cell count and internal standard, and Pareto scaling.

**Synovial biopsy study**. Within the framework of another research study[43] (manuscript in preparation), an explorative genome-wide transcriptional profiling study was performed on synovial tissue biopsies obtained from 13 autoantibody-positive (IgM rheumatoid factor and/or anti-citrullinated protein antibody) arthralgia patients using Agilent arrays (Agilent technologies, Amstelveen, The Netherlands). The study was approved by the institutional review board of the Academic Medical Center (Amsterdam, The Netherlands), and all study subjects gave written informed consent. At the time of synovial biopsy of a knee joint by mini-arthroscopy these individuals did not display any evidence of arthritis as assessed by an experienced rheumatologist. However, they are "at risk" for developing RA and therefore prospectively followed over time to study the possible development of arthritis[42]. Expression levels of genes of interest were extracted from this dataset and investigated for their possible association with arthritis development. To this end, RA-risk individuals were stratified based on their mRNA levels into relative low or high expressers compared to the median values of the whole population, after which arthritis-free survival was compared to each other using a log-rank (Mantel–Cox) test.

**Statistical analyses**. For flow cytometry, ELISA and qPCR data, statistical analyses were performed in GraphPad Prism. First, normality of data distribution of either raw or transformed data was analysed by performing a D'Agostino and Pearson omnibus normality test. Normally distributed data was compared with repeated-measures one-way analysis of variance test with post hoc Dunnett's correction for multiple comparisons to cells with no treatment. When data were not normally distributed, a Friedman test with post hoc Dunn's multiple comparison correction was applied. When the $N$ was too small to pass a normality test ($n < 8$), a non-parametric test was performed and post hoc power calculations were performed to confirm significance. Differences were considered significant at the 95% confidence level.

**Microarray data analysis**. Raw data were summarised and normalised using Robust Multi-array Average. Probes without an annotated HUGO gene symbol were removed from the analysis. Probes with a normalised standard deviation ($\sigma/\sigma_{max}$) lower than 0.05 were removed from the analysis. Differential expression comparisons between the IFN$\gamma^+$IL-10$^-$, IFN$\gamma^+$IL-10$^+$ and IFN$\gamma^-$IL-10$^+$ populations was computed using Limma taking into account the correlation between samples from the same subject. No variance filter was applied. The following model was fit: "'expr ~ population + sex'". For each comparison of interest, contrasts were extracted using the makeContrasts function in the R package Limma. $P$ values were corrected for multiple testing across all genes using the Benjamini–Hochberg procedure. Adjusted $P$ values lower than 0.05 were considered significant. Clustering analysis was performed using $k$-means (as implemented in the R package stat) on IFN$\gamma^+$IL-10$^-$, IFN$\gamma^+$IL-10$^+$ and IFN$\gamma^-$IL-10$^+$ populations. The number of clusters was set to three and the number of random initialisations was set to 10. All probe sets were used for clustering. PCA was performed using the R function prcomp from the package stat and the first two components were used to visualise the results of the clustering algorithm. Plots were created using the R package ggplot2. Correlation between IL-10 protein expression and probe expression was estimated using linear regression adjusting for gender. $P$ values were corrected for multiple testing using the Benjamini–Hochberg procedure. Corrected $P$ values lower than 0.05 were considered significant. Pathway analysis was performed on the list of significantly correlated probes using Ingenuity Pathway Analysis. For Jurkat T cell microarray analyses, results were compiled using Genome Studio (illumine) where they were normalised using quantile normalisation and pathway analyses performed as above.

## Data availability

The data generated for this work have been deposited at Gene Expression Omnibus (GEO) under the accession codes GSE119416 and GSE122499. All other data supporting the findings of this study that are not included in the figures or the supplementary information of this paper are available from the corresponding authors upon reasonable request.

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

## Acknowledgements

This work was supported by the IMI-funded project BeTheCure, 115142-2 (E.P., L.G.M. v.B. and A.P.C.) and from the EU/EFPIA Innovative Medicines Initiative 2 Joint Undertaking RTCure grant no. 777357 (E.P. and A.P.C.); the National Institute for Health Biomedical Research Centre, at Guy's and St Thomas' NHS Foundation Trust and King's College London (J.A.B.); the King's Bioscience Institute and the Guy's and St. Thomas' Charity Prize PhD programme in Biomedical and Translational Science (C.A.R.); the Medical Research Council (Lipid Profiling and Signalling, MC UP A90 1006 and Lipid Dynamics and Regulation, MC PC 13030 (Z.H. and J.L.G.); Arthritis Research UK (L.S.T., programme grant 21139); the Division of Intramural Research, National Heart, Lung, and Blood Institute, National Institutes of Health (C.K.) and the BBSRC (BBK019112/1) and EU-Welsh government Ser Cymru programme (P.G.). This work was also supported by infrastructure funded by the National Institute for Health Research (NIHR) Clinical Research Facility and Biomedical Research Centre at Guy's and St. Thomas' NHS Foundation Trust and King's College London (reference: guysbrc-2012-17). The content is solely the responsibility of the authors and does not necessarily represent the official views of the National Institutes of Health. The authors thank Harriet Purvis, Fiona Clarke, Cristina Sanchez-Blanco, Georgina Cornish and Tamlyn Peel for critical discussions.

## Author contributions

E.P. designed, performed and analysed experiments and co-wrote the manuscript; W.W. performed experiments; J.A.B., C.A.R., G. L. and L.G.M.v.B performed experiments and participated in manuscript preparation; R.M. performed bioinformatics analyses and contributed to figure preparations; Z.H. and J.L.G. performed mass spectrometry experiments; P.L. performed the CYT-1/CYT-2 microarray experiments; K.S.F. analysed experiments and participated in manuscript preparation; J.G.G. participated in manuscript preparation; E.d.R. supervised bioinformatics analyses; K.A.R., L.S.T., C.K. and P.G. contributed to design and edited the manuscript; A.P.C. conceived the project, designed experiments and co-wrote the manuscript.

## Additional information

**Competing interests:** K.S.F. is an employee of Novo Nordisk; J.G.G. is an employee of Eli Lilly. The remaining authors declare no competing interests.

