## [Peer Review File · Nature Communications]

Reviewers' comments:

Reviewer #1 (Remarks to the Author):

This study follows up on the finding that CD46 cross-linking on human T cells cultured under Th1 conditions promotes IL10 expression with experiments to define the mechanism involved. An examination of gene expression differences in the IL10 IFN γ double expressing versus single expressing cells reveals cholesterol biosynthetic pathway genes as being differentially expressed. Treatment with statins is then shown to diminish IL10 induction in this culture system and addition of the downstream intermediate mevalonate, but not isoprenoid precursors, restores IL10 expression. IL4 expression may also be affected whereas IFN γ , IL17 and TNF are not, at least under the culture conditions shown. 25-HC, a repressor of SREBPs, also causes a reduction in the frequency of IFN γ +IL10+ cells in the culture system. These data provide support for the conclusion that the cholesterol biosynthetic pathway is somehow needed for the generation of human IL10-producing cells in vitro. Data with a small number of human RA patients is used to suggest that this pathway may be active in vivo. Showing a differential requirement for the CBP in certain effector T cell states is of significance even if the mechanism for this effect is not yet defined. However, there are some specific concerns, as follows:

1. The heat map in Fig. 1b is confusing. It is unclear what is being plotted, in terms of whether the data are somehow related to IL10 protein MFI (multiple sorts at different IL10 levels)? It would be more straightforward to show 3 way supervised clustering of the IL10+IFN γ +, IL10-IFN γ +, and IL10-IFN γ - populations. Also, the data in Supplementary Table 1 show the CBP genes as having a negative fold change. This table needs to be more clearly annotated.
2. PCR confirmation of some of the core CBP genes should be shown.
3. Cholesterol biosynthesis is important for cell proliferation. It seems possible that to reach an IL10 producing state the Th1 type cells may need to proliferate more than cells that have not reached that state. For example, it has been reported that IL10 producers are the most terminally differentiated T cells (e.g. Saraiva et al., 2009 Immunity 31, 209). The authors make one comment in the discussion that there were not 'discernible effects on proliferation'. However, no proliferation data (such as CFSE dilution profiles for cells cultured \pm statin and \pm 25-hydroxycholesterol) are shown.
4. The study largely relies on an anti-CD46 based method of promoting IL10 producing cell induction. Fig 1g shows data for one additional approach, culturing cells with anti-TNF. However, the data for this approach show less evidence of a titration effect and the figure lacks a summary panel to show whether the effect was statistically significant. It is also important to know if 25-HC represses IL10 expression in this (or some other) alternative culture system.

Minor

1. It is unusual to have an entire results figure (Fig. 2) taken up by a diagram of an established pathway (without any actual data).
2. Reference 28 is cited as indicating that 25-HC augmented IL17 cell numbers when the study appeared to reach the opposite conclusion.
3. It should be noted that U18666A also inhibits NPC1 (Lu et al., 2015 eLIFE e12177).

Reviewer #2 (Remarks to the Author):

In the manuscript "The cholesterol biosynthesis pathway regulates IL-10 expression in human Th1 effector cells," Perucha et al. report new insights into the role of cholesterol synthesis in the regulation of Th1 switch to IL-10 phenotype, suggesting that cholesterol flux is required for the secretion of IL-10 in Th1 cells, and proposing that the expression of 25-hydroxycholesterol and several cholesterol metabolism enzymes can be used as predictors for the development of rheumatoid arthritis. The authors used global gene expression profiling and pharmacological

inhibitors against cholesterol related metabolic enzymes combined with rescue experiments to dissect the mechanism by which cholesterol synthesis regulates Th1 IL-10 phenotype switch. The work is unique in shedding some light on the effect of statins on the immune system and related controversy. The findings here are interesting and potentially important to our understanding of cellular metabolic influences on immune cell function.

I have the following concerns:

1. The authors show that T cells activated with anti-CD3 and anti-CD46 have decreased production of the anti-inflammatory cytokine IL-10 when cholesterol synthesis is blocked using atorvastatin. This is also true of CD3/CD46-stimulated T cells treated with 25-hydroxycholesterol. However, the authors need to better explain the physiologic relevance of T cells activated with anti CD3/CD46 (as opposed to anti-CD3/CD28). It is not apparent that these are physiologically relevant cells.
2. I have concerns about the use of transfected Jurkat cells in Figure 1, as these leukemic T cells are transformed and therefore have a very different metabolic state than primary T cells.
3. Authors should show effects of atorvastatin on T cell proliferation in their model, in addition to survival data.
4. Do these findings simply relate to the fact that blocking cholesterol biosynthesis prevents cell division or proliferation because the cells lack the means to make lipid membranes? And if so, are there other immune cell activation models in which we would find similar results?
5. Human data observations are compelling, but do not show mechanism. Authors should show the effects of disrupting cholesterol metabolism in a mouse model of inflammation *in vivo*.
6. There are concerns with how the flow cytometry data are normalized to 100% in the non-treated group. If every mouse in that group is normalized to 100%, this calls into question how the other experimental points are compared to the control mice. The authors should instead graph both absolute (not normalized) % cells, as well as # of cells, that are identified as IFN-g +/- IL-10 producing in Figure 1, and use similar data reporting for other flow cytometry figures in the manuscript.
7. For Supp Figure 2, what does CD69 and CD25 production usually look like in the anti-CD3/CD46 treated group, and how does that compare to CD69 and CD25 production in CD3/CD28 activated cells?

Reviewer #3 (Remarks to the Author):

The manuscript by Perucha and colleagues examines the influence of the cholesterol biosynthetic pathway on the induction of IL-10 producing Th1 cells in humans. The authors initially show significant enrichment for sterol metabolism genes in IL-10 producing T cells when compared to IFN γ producing counterparts. They subsequently demonstrate that pharmacologic inhibition of the mevalonate pathway specifically impacts the production of IL-10, but does not grossly alter IFN γ or IL-17 production. Importantly the addition of mevalonic acid restores IL-10 production, indicating that metabolic flux through this pathway is necessary for IL-10 production in this system. The authors then pharmacologically dissect the metabolic pathways impacting IL-10 production downstream of HMGCR using pharmacologic systems. The product of these studies rule out a number of pathways, including the non-steroidal branches and vitamin D3 metabolism. Additionally, the effect appears to be dependent on flux through the distal branch of the cholesterol biosynthetic pathway. The authors then replicate their findings using 25-HC, a sterol metabolite which inhibits the cholesterol biosynthetic pathway, further suggesting a critical role for

this pathway in regulating the transition to IL-10 producing cells. Finally, the authors correlate arthritis disease severity with expression of the cholesterol biosynthetic genes and Ch25h.

Overall, this is an interesting story which highlights the role of sterol metabolism with T cell function. Unfortunately, while the authors demonstrate an importance of the cholesterol biosynthetic pathway on the induction of IL-10, they cannot pinpoint a specific metabolite or group of metabolites that are responsible for this effect. Thus, they must rely on "metabolic flux" of the pathway to mechanistically explain their observations. Furthermore, the authors provide no evidence as to how cholesterol biosynthetic flux specifically regulates IL-10 without impacting other cytokines. Without identifying the metabolite that is regulating this transition, or demonstrating a protein that monitors flux (and by extension regulates IL-10 transcription), the manuscript becomes another example of how pharmacologic perturbations of cholesterol metabolism impacts T cell function. Interesting, but not a significant mechanistic advance.

Major questions/issues are:

- 1) A prediction of the authors data is that replenishing biosynthetic intermediates of the distal branch of synthesis should restore IL-10 production. This should be done and compared to addition of MBCD-cholesterol.
- 2) The authors discuss "flux" quite a bit, but provide no direct evidence of changes in flux. This would be important given that their MS data on statins shows no change in total cholesterol and CEs (a somewhat surprising effect).
- 3) The authors are lacking any mechanism linking sterol metabolism to IL-10 production. Does the perturbation in cholesterol homeostasis impact IFN gamma receptor signaling or trafficking? Is IL-10 production dependent on IFN γ signaling?

Minor issues

Fig. 1. Can the authors provide some kinetic studies to demonstrate when IL-10 comes up relative to IFN γ in this system and how does statins impact this?

Fig 4. It is not clear what point the authors are trying to make with the pravastatin. This should be clarified.

Fig. 5. The imaging of SREBP2 is unconvincing and conventional western blots on nuclear lysates should be provided.

Figure 5. Do these cells make 25HC? If so, is CH25HC enriched in one of the populations?

Fig. 6. The authors have previously shown that IFN γ signaling regulate Ch25h and genes of the cholesterol biosynthetic pathway. So, are they simply showing that these genes are surrogate biomarkers for IFN γ expression/function in the disease?

REVIEWERS' COMMENTS

Reviewer #1 (Remarks to the Author):

This study follows up on the finding that CD46 cross-linking on human T cells cultured under Th1 conditions promotes IL10 expression with experiments to define the mechanism involved. An examination of gene expression differences in the IL10 IFN γ double expressing versus single expressing cells reveals cholesterol biosynthetic pathway genes as being differentially expressed. Treatment with statins is then shown to diminish IL10 induction in this culture system and addition of the downstream intermediate mevalonate, but not isoprenoid precursors, restores IL10 expression. IL4 expression may also be affected whereas IFN γ , IL17 and TNF are not, at least under the culture conditions shown. 25-HC, a repressor of SREBPs, also causes a reduction in the frequency of IFN γ +IL10+ cells in the culture system. These data provide support for the conclusion that the cholesterol biosynthetic pathway is somehow needed for the generation of human IL10-producing cells in vitro. Data with a small number of human RA patients is used to suggest that this pathway may be active in vivo. Showing a differential requirement for the CBP in certain effector T cell states is of significance even if the mechanism for this effect is not yet defined. However, there are some specific concerns, as follows:

1. The heat map in Fig. 1b is confusing. It is unclear what is being plotted, in terms of whether the data are somehow related to IL10 protein MFI (multiple sorts at different IL10 levels)? It would be more straightforward to show 3 way supervised clustering of the IL10+IFN γ +, IL10-IFN γ +, and IL10+IFN γ - populations. Also, the data in Supplementary Table 1 show the CBP genes as having a negative fold change. This table needs to be more clearly annotated.

In Fig. 1b differentially expressed genes are plotted against protein levels, derived from the MFI IL-10 parameter, and presented as a ratio between IFN γ ⁺IL-10⁺ and IFN γ ⁻IL-10⁻ negative populations. This takes into consideration inter-experimental variability in IL-10 levels between donors. This has been clarified in the results section (page 5 of the manuscript).

A 3-way unsupervised analysis plot has now been incorporated into the manuscript (Supplementary Fig. 1c), together with a more detailed explanation of the analyses performed. Thank you for suggesting this.

Finally, the results section highlights that the relationship between the genes belonging to the CBP and IL-10 protein levels is inverse. This is also stated in Fig. 1 legend and explained in the text. The negative “r” value highlighted in Supplementary Table 1 is in agreement with this. We hope that this now provides sufficient clarity.

2. PCR confirmation of some of the core CBP genes should be shown.

This is a good suggestion. These data are now described in the results section, and have been added to Supplementary Fig. 1d.

3. Cholesterol biosynthesis is important for cell proliferation. It seems possible that to reach an IL10 producing state the Th1 type cells may need to proliferate more than cells that have not reached that state. For example, it has been reported that IL10 producers are the most

terminally differentiated T cells (e.g. Saraiva et al., 2009 Immunity 31, 209). The authors make one comment in the discussion that there were not 'discernible effects on proliferation'. However, no proliferation data (such as CFSE dilution profiles for cells cultured ±statin and ±25-hydroxycholesterol) are shown.

Thank you for highlighting this point. We have performed additional proliferation experiments and were not able to detect cell division at the 36h experimental time point (Supplementary Fig. 3d). We believe that this fairly conclusively rules out the possibility that the profound reductions in IL-10 observed at this early time point are due to disruption in cellular proliferation. This data complements our observations demonstrating that statin treated T cells do not show detectable changes in total cholesterol cell content (Fig. 3f), which might conceivably have influenced the capacity of cells to divide.

4. The study largely relies on an anti-CD46 based method of promoting IL10 producing cell induction. Fig 1g shows data for one additional approach, culturing cells with anti-TNF. However, the data for this approach show less evidence of a titration effect and the figure lacks a summary panel to show whether the effect was statistically significant. It is also important to know if 25-HC represses IL10 expression in this (or some other) alternative culture system.

We have undertaken more experiments to address this. To further support the concept that blockade of cholesterol pathway with atorvastatin regulates IL-10 expression in contexts besides CD46, we provide new evidence on other IL-10-inducing *in vitro* systems such as anti-TNF α (Fig. 2g and Supplementary Fig. 3h) and IFN α (Supplementary Fig. 3i). Moreover, we show the same effect on IL-10 regulation in the presence of 25-HC (Supplementary Fig. 5 b and c). All together, these data show that cholesterol biosynthesis blockade downregulated IL-10 expression in 3 different *in vitro* culture conditions.

Minor

1. It is unusual to have an entire results figure (Fig. 2) taken up by a diagram of an established pathway (without any actual data).

The pathway schematic has been moved to supplementary information (Supplementary Fig. 2).

2. Reference 28 is cited as indicating that 25-HC augmented IL17 cell numbers when the study appeared to reach the opposite conclusion.

Thank you for pointing this out. Given that the role for IL-17 is not the focus of our study we have removed this reference.

3. It should be noted that U18666A also inhibits NPC1 (Lu et al., 2015 eLIFE e12177).

Thank you. This has now been included in the manuscript.

Reviewer #2 (Remarks to the Author):

In the manuscript "The cholesterol biosynthesis pathway regulates IL-10 expression in human Th1 effector cells," Perucha et al. report new insights into the role of cholesterol synthesis in the regulation of Th1 switch to IL-10 phenotype, suggesting that cholesterol flux

is required for the secretion of IL-10 in Th1 cells, and proposing that the expression of 25-hydroxycholesterol and several cholesterol metabolism enzymes can be used as predictors for the development of rheumatoid arthritis. The authors used global gene expression profiling and pharmacological inhibitors against cholesterol related metabolic enzymes combined with rescue experiments to dissect the mechanism by which cholesterol synthesis regulates Th1 IL-10 phenotype switch. The work is unique in shedding some light on the effect of statins on the immune system and related controversy. The findings here are interesting and potentially important to our understanding of cellular metabolic influences on immune cell function.

I have the following concerns:

1. The authors show that T cells activated with anti-CD3 and anti-CD46 have decreased production of the anti-inflammatory cytokine IL-10 when cholesterol synthesis is blocked using atorvastatin. This is also true of CD3/CD46-stimulated T cells treated with 25-hydroxycholesterol. However, the authors need to better explain the physiologic relevance of T cells activated with anti CD3/CD46 (as opposed to anti-CD3/CD28). It is not apparent that these are physiologically relevant cells.

More information regarding the relevance of CD46 has now been included in the introduction (page 3 of the manuscript).

2. I have concerns about the use of transfected Jurkat cells in Figure 1, as these leukemic T cells are transformed and therefore have a very different metabolic state than primary T cells.

The aim of the Jurkat T cell experiments shown in Fig.1 was not so much to study an alternative T cell substrate with a comparable metabolic state, but to exploit a commonly adopted experimental T cell model to demonstrate unambiguously that a switch in the CD46 cytoplasmic tail is sufficient to induce a signature indicative of dysregulated cholesterol biosynthesis, similar to that observed in primary human T cells. Regarding the metabolic state of Jurkat cells it should be noted that data from our own laboratory has suggested that this cell line mimics primary cells in terms of glycolysis and oxidative phosphorylation (Kolev et al *Immunity* 2015; 42:1033-47).

3. Authors should show effects of atorvastatin on T cell proliferation in their model, in addition to survival data.

As discussed above in response to the similar comment from reviewer 1, we have now performed proliferation experiments in which we were not able to detect cell division at the 36h experimental timepoint (Supplementary Fig. 3d), ruling out the possibility that the profound reduction in IL-10 observed is due to disturbances in cellular proliferation.

4. Do these findings simply relate to the fact that blocking cholesterol biosynthesis prevents cell division or proliferation because the cells lack the means to make lipid membranes? And if so, are there other immune cell activation models in which we would find similar results?

Our data show that atorvastatin treatment does not affect cell viability (Supplementary Fig. 3b and 3c) nor total cholesterol cell content (Fig. 3f). Together with the new proliferation data generated (Supplementary Fig. 3d), we believe that we have demonstrated that the IL-10

effect is independent of cell proliferation, and that IL-10 production is uncoupled in the absence of detectable changes in total cellular cholesterol content.

We have also provided evidence that cholesterol pathway-mediated IL-10 regulation is also relevant in other *in vitro* CD4⁺ T cell inducing systems (Fig. 2g; Supplementary Fig. 3h and I and Supplementary Fig. 5 b and c). Finally, we have conclusive evidence that cholesterol biosynthesis regulates IL-10 expression in primary human B cells; however, this is the topic of a separate study (manuscript in preparation).

5. Human data observations are compelling, but do not show mechanism. Authors should show the effects of disrupting cholesterol metabolism in a mouse model of inflammation in vivo.

To provide new mechanistic insights we have performed experiments revealing that disrupting cholesterol metabolism regulates the expression of c-Maf, one of the master transcription factors of IL-10 gene transcription in CD4⁺ T cells. These new data are summarised in Fig. 5. The contribution of cholesterol metabolism to T cell immunity, as well as the impact of pharmacological intervention (eg with statins) is not well understood, and may be different to experiments described in man. However, we were reassured of our findings by the work of Caroline Pot et al, after our work was first submitted for review, who reported that IL-10 expression in IL-27 induced murine Tr1 cells is also regulated by oxysterols *in vitro* and *in vivo*. Interestingly, Vigne *et al* demonstrate that IL-27 induces expression of *Ch25h* the gene encoding the enzyme required for biosynthesis of 25-HC in CD4⁺ T cells. We have cited these new data in the revised manuscript. We also place the data in context of our own *in vivo* human model of inflammation, in which we demonstrate that expression of a number of genes that underpin cholesterol biosynthesis predict progression of high risk individuals (a very early inflammatory state) to developing rheumatoid arthritis (a fully established inflammatory state). This *in vivo* “human model” is highly relevant to pathways of resolution.

6. There are concerns with how the flow cytometry data are normalized to 100% in the non-treated group. If every mouse in that group is normalized to 100%, this calls into question how the other experimental points are compared to the control mice. The authors should instead graph both absolute (not normalized) % cells, as well as # of cells, that are identified as IFN-g +/- IL-10 producing in Figure 1, and use similar data reporting for other flow cytometry figures in the manuscript.

We highlight that our manuscript does not contain any data derived from experiments performed in the mouse, as the reviewer is suggesting. All human T cell data have been normalised in the manuscript to account for donor variability, and experiments undertaken over extended periods of time. However, please note that all statistical analyses have been performed on the raw data. We are happy to provide the raw data files as supplementary information if requested.

7. For Supp Figure 2, what does CD69 and CD25 production usually look like in the anti-CD3/CD46 treated group, and how does that compare to CD69 and CD25 production in CD3/CD28 activated cells?

Representative dot-plots of flow cytometric expression data have been incorporated in supplementary information (Supplementary Fig. 3e), including data from additional experiments (Supplementary Fig. 3f). A CD3/CD28 comparison is shown below to address

this reviewer's concerns about the possible differences between CD46 and CD28 activation. T cell activation is comparable when using CD69 and CD25 expression as readouts.

Reviewer #3 (Remarks to the Author):

The manuscript by Perucha and colleagues examines the influence of the cholesterol biosynthetic pathway on the induction of IL-10 producing Th1 cells in humans. The authors initially show significant enrichment for sterol metabolism genes in IL-10 producing T cells when compared to IFN γ producing counterparts. They subsequently demonstrate that pharmacologic inhibition of the mevalonate pathway specifically impacts the production of IL-10, but does not grossly alter IFN γ or IL-17 production. Importantly the addition of mevalonic acid restores IL-10 production, indicating that metabolic flux through this pathway is necessary for IL-10 production in this system. The authors then pharmacologically dissect the metabolic pathways impacting IL-10 production downstream of HMGCR using pharmacologic systems. The product of these studies rule out a number of pathways, including the non-steroidal branches and vitamin D3 metabolism. Additionally, the effect appears to be dependent on flux through the distal branch of the cholesterol biosynthetic pathway. The authors then replicate their findings using 25-HC, a sterol metabolite which inhibits the cholesterol biosynthetic pathway, further suggesting a critical role for this pathway in regulating the transition to IL-10 producing cells. Finally, the authors correlate arthritis disease severity with expression of the cholesterol biosynthetic genes and Ch25h.

Overall, this is an interesting story which highlights the role of sterol metabolism with T cell function. Unfortunately, while the authors demonstrate an importance of the cholesterol biosynthetic pathway on the induction of IL-10, they cannot pinpoint a specific metabolite or group of metabolites that are responsible for this effect. Thus, they must rely on "metabolic flux" of the pathway to mechanistically explain their observations. Furthermore, the authors provide no evidence as to how cholesterol biosynthetic flux specifically regulates IL-10 without impacting other cytokines. Without identifying the metabolite that is regulating this transition, or demonstrating a protein that monitors flux (and by extension regulates IL-10 transcription), the manuscript becomes another example of how pharmacologic

perturbations of cholesterol metabolism impacts T cell function. Interesting, but not a significant mechanistic advance.

Thank you for highlighting the fact that this is an interesting story. We are pleased to address further the concerns of this reviewer below, highlighting the mechanistic advances that we now report in our revised manuscript.

Major questions/issues are:

1) A prediction of the authors data is that replenishing biosynthetic intermediates of the distal branch of synthesis should restore IL-10 production. This should be done and compared to addition of MBCD-cholesterol.

This is a great suggestion – thank you. We have now performed experiments with cultured cells either in media containing delipidised serum to disrupt cholesterol flux (lipid free media, LFM), as well as media supplemented with cholesterol to normalise cholesterol flux when inhibited in the presence of 25-HC. These data, which support our model conclusively, have now been incorporated in the manuscript (Fig. 4e; Supplementary Fig. 5a and Fig. 5e).

2) The authors discuss “flux” quite a bit, but provide no direct evidence of changes in flux. This would be important given that their MS data on statins shows no change in total cholesterol and CEs (a somewhat surprising effect).

We agree that this is an important issue, and thank the reviewer for highlighting this point. We have now undertaken mass spectrometry experiments to provide an unbiased and systematic lipidomic analysis of T cells stimulated in the presence or absence of atorvastatin (Supplementary Fig. 4e and Supplementary Table 3). The results highlight the specificity of the link between cholesterol metabolism and IL-10, given that no other lipid family was altered by the treatment. Atorvastatin has already been shown to alter T cell biology without affecting cholesterol levels, even at a higher doses and longer exposure times used in our experiment (see Blank et al 2007, *Journal of Immunology*).

3) The authors are lacking any mechanism linking sterol metabolism to IL-10 production. Does the perturbation in cholesterol homeostasis impact IFN gamma receptor signaling or trafficking? Is IL-10 production dependent on IFN γ signaling?

Please see our response to reviewer 2, regarding the mechanistic role of c-Maf. These data are now incorporated into Fig. 5.

We have also incorporated in our introduction a more comprehensive and accessible description of the human Th1 switching model (which has so far no observed in this form in mice), citing our extensive published analyses of the initial requirements for IFN γ production to generate autocrine IL-10 and initiate the safe human Th1 shut down programme – based on published work demonstrating that CD46 deficient individuals lack IL-10 due to a complete lack of IFN γ response (but can produce other cytokines well) and that dysregulation of CD46-mediated signals contribute to hyperactive Th1 responses (uncontrolled IFN γ production with reduced IL-10 switching) in patients with rheumatoid arthritis and systemic lupus erythematosus.

Minor issues

Fig. 1. Can the authors provide some kinetic studies to demonstrate when IL-10 comes up relative to IFN γ in this system and how does statins impact this?

Details of the kinetics of switching have been reported by our lab (Cardone et al 2010, *Nature Immunology*) and are now included in the text. We have also undertaken detailed kinetics of Th switching with statin treatment. IL-10 protein is detectable at 12h, with a peak of expression at 36h time point (data shown below for 3 independent donors). Atorvastatin inhibits this upregulation from the 12h time point, although the effect is more prominent at 36h. In contrast, the levels of intracellular IFN γ remained unchanged. We have also analysed mRNA levels at earlier time points, but IL-10 mRNA is barely detectable.

Fig. 4. It is not clear what point the authors are trying to make with the pravastatin. This should be clarified.

This point has now been clarified in the text (page 6 of the manuscript).

Fig. 5. The imaging of SREBP2 is unconvincing and conventional western blots on nuclear lysates should be provided.

For lack of clarity, these data have now been removed from the manuscript.

Figure 5. Do these cells make 25HC? If so, is CH25HC enriched in one of the populations?

Bulk murine CD4⁺ T cells do make oxysterols upon stimulation (Vigne et al 2017, *Frontiers in Immunology*). [Redacted] In light of this, we have also performed detailed experiments aimed at quantifying *CH25H* by qPCR under different conditions of stimulation in human CD4⁺ T cells; the kinetics of expression are very similar to those reported in macrophages by our group (Blanc *et al* 2013, *Immunity*). [Redacted]

[Figure redacted]

To the best of our knowledge, there are no reports providing evidence that IFN γ induces *CH25H* expression in mouse or human systems. We wonder if the reviewer is referring to the induction of oxysterol biosynthesis by type I IFNs? This has certainly been published by our group (Blanc *et al* 2013, *Immunity*). The context here would be that type I IFNs, which induce IFN-induced expression signatures in a wide variety of autoimmune mediated inflammatory diseases, would play a key role in inducing a non-permissive IL-10 state though induction of *CH25H* that would support anti-viral immunity but would be detrimental in terms of resolution of inflammation.

Reviewers' comments:

Reviewer #1 (Remarks to the Author):

With their revisions the authors have minimally but adequately addressed my earlier concerns.

It is unfortunate that the authors did not underline their changes to facilitate the re-review process.

I don't understand the author's response to reviewer 3's comment that 'there are not reports providing evidence that IFN γ regulates CH25H expression in mouse or human systems'. The paper they cite (Blanc/Ghazal et al 2013 Immunity) shows data in several figures regarding IFN γ -mediated induction of Ch25h mRNA. This error is especially surprising given that Ghazal is a coauthor.

Define adalimumab

Fig 3d legend should say U18666A

The source of calcitriol needs to be indicated.

Suppl. Fig. 5 legend was unreadable.

Editorial note: Reviewer #2 declined to re-review.

Reviewer #3 (Remarks to the Author):

The revised manuscript by Cope and colleagues examines the influence of the cholesterol biosynthetic pathway on the induction of IL-10 producing Th1 cells in humans. In this revision, the authors have reasonably addressed my concerns, and should be congratulated on a very nice series of studies.

Minor issues:

1) In my copy of the manuscript, figure S5 was missing half of the figure. This could be a pdf conversion issue, but if not, this should be addressed.

2) It is also surprising that addition of cholesterol restores the expression of SREBP2 target genes in 25HC treated cells (Supplementary Fig. 5a). The addition of exogenous cholesterol should also decrease CBP flux, and decrease SREBP processing/transcriptional activity. The authors conclude that "supplementation of 25-HC treated cultures with cholesterol reversed this downregulation (Supplementary Fig. 5a), confirming that 25-HC perturbs CBP homeostasis in CD4+ T cells." I am not certain that the addition of cholesterol proves this, and the authors might want to reconsider their interpretation of this data. This is not a major issue, but the authors might want to reconsider how they interpret these data.

###Mediating comments for previous Reviewer #2 comments###

Please find the requested assessment of reviewer #2's concerns for "The cholesterol biosynthesis pathway regulates IL-10 expression in human Th1 effector T cells" by Cope and colleagues. The authors address the majority of the review#2's concerns. Specifically, they provide additional introduction on the relevance of CD3/46 activation model. They also demonstrate that atorvastatin treatment does not interfere with proliferation and viability of their cells in this system. They also show that atorvastatin treatment does not alter total cellular cholesterol levels.

There was a request by this reviewer for additional mechanistic studies linking cholesterol homeostasis with IL-10 production. This reviewer requested a mouse model of disease. The

revised manuscripts now provide some correlative evidence linking IL-10 with cholesterol via control of c-MAF expression. This is a modest advance in mechanism, and I am not sure that it would have satisfied the reviewer's concerns. I defer to the editor on this issue.

All other minor concerns appear to be addressed.

Reviewer #4 (Remarks to the Author):

This is a very novel paper concerning a major medical issue. However, I will comment only on the statistical issues.

Specific Comments:

1. Line 365: IQR= (2,44) is too wide, which tells that the variability is too much. This implies that the sample sizes are not large enough.
2. Line 371: Confidence Interval, CI should be written as (1.0, 27.9) which is also very wide.
3. Lines 645 to 653: The methodology is described well but for the nonparametric test with $n < 8$, I wonder what will be the power of the test? The authors should include a power analysis.
4. Line 669: What is the justification of choosing three clusters.
5. Line 676: Authors should also report FDR (False Discovery Rate) as well as FP (False Positive) and FN (False Negative).

General Comments:

There are too much emphasizes are given on p-values.

In view of the prevalent misuses of and misconceptions concerning p-values, some statisticians prefer to supplement or even replace p-values with other approaches. These include methods that emphasize estimation over testing, such as credibility, or prediction intervals; Bayesian methods; alternative measures of evidence, such as likelihood ratios or Bayes Factors; and other approaches such as decision-theoretic modeling and false discovery rates. I suggest authors to add few other measures.

13th August 2018

To the reviewers of the manuscript **NCOMMS-17-16170A-Z**

In further support of the revision of our manuscript entitled “*The cholesterol biosynthesis pathway regulated IL-10 expression in human Th1 effector T cells*”, we have now addressed the latest comments with a minor revision to the main manuscript (with tracked changes) and a detailed point-by-point responses to each reviewers’ critique (shown in *italics*) shown below. We hope you find our revised document and our responses favourable.

With best wishes,

Yours sincerely,

Esperanza Perucha and Andrew Cope

REVIEWERS' COMMENTS

Reviewer #1 (Remarks to the Author):

With their revisions the authors have minimally but adequately addressed my earlier concerns. It is unfortunate that the authors did not underline their changes to facilitate the re-review process.

We understand this reviewer's frustration, but on the recommendation of the editor we were invited to submit a new manuscript rather than a revised one, and so at the time did not think that underlining changes was appropriate.

I don't understand the author's response to reviewer 3's comment that 'there are not reports providing evidence that IFN γ regulates CH25H expression in mouse or human systems'. The paper they cite (Blanc/Ghazal et al 2013 Immunity) shows data in several figures regarding IFN γ -mediated induction of Ch25h mRNA. This error is especially surprising given that Ghazal is a coauthor.

Thank you for pointing out this error. The reviewer is quite correct in stating that IFN γ regulates CH25H gene expression, as described in Blanc *et al* (Fig 2E), as well as by type I IFN signalling (Fig 6). We surmise that in pre-RA synovium both type I and type II IFN signalling are contributing to the induction of these genes. We cannot rule out, however, the possibility that the metabolic pathways are the primary drivers of the effector response, as opposed to IFNs. This is an area of current investigation.

Define adalimumab.

Adalimumab has now been defined in the text (page 7).

Fig 3d legend should say U18666A

The legend to Fig 3 has been corrected.

The source of calcitriol needs to be indicated.

The source was already indicated in the methods section (Sigma), but calcitriol was named as 1 α ,25-dihydroxyvitamin D3. Methods have now been amended for clarification (page 16).

Suppl. Fig. 5 legend was unreadable.

Thank you for pointing this out. We think this may have been a technical issue during file upload, so we will make sure all the files are correctly uploaded during re-submission.

Reviewer #3 (Remarks to the Author):

The revised manuscript by Cope and colleagues examines the influence of the cholesterol biosynthetic pathway on the induction of IL-10 producing Th1 cells in humans. In this revision, the authors have reasonably addressed my concerns, and should be congratulated on a very nice series of studies.

Minor issues:

1) In my copy of the manuscript, figure S5 was missing half of the figure. This could be a pdf conversion issue, but if not, this should be addressed.

Thank you for pointing this out. We will make sure all the files are correctly uploaded.

2) It is also surprising that addition of cholesterol restores the expression of SREBP2 target genes in 25HC treated cells (Supplementary Fig. 5a). The addition of exogenous cholesterol should also decrease CBP flux, and decrease SREBP processing/transcriptional activity. The authors conclude that “supplementation of 25-HC treated cultures with cholesterol reversed this downregulation (Supplementary Fig. 5a), confirming that 25-HC perturbs CBP homeostasis in CD4+ T cells.” I am not certain that the addition of cholesterol proves this, and the authors might want to reconsider their interpretation of this data. This is not a major issue, but the authors might want to reconsider how they interpret these data.

Cholesterol suppresses its own biosynthesis via inhibition of SREBP-2 translocation to the Golgi by interacting with SCAP. However, oxysterols are more potent inhibitors of this metabolic pathway (via interaction with INSIG proteins). Our work shows that in the presence of 25-HC, supplementation with cholesterol is able to restore, at least partially, the levels of key enzymes and proteins in the cholesterol pathway (Suppl Fig. 5a), together with IL-10 levels. Based on these data alone, we can state that the perturbation in CBP induced by 25-HC is restored in the presence of cholesterol, and that the mechanisms that are likely to be linked to SREBP-2 functions, hypothesis that we are currently testing. The text has been amended accordingly (see page 10).

Mediating comments for previous Reviewer #2 comments

Please find the requested assessment of reviewer #2’s concerns for “The cholesterol biosynthesis pathway regulates IL-10 expression in human Th1 effector T cells ” by Cope and colleagues.

The authors address the majority of the review#2’s concerns. Specifically, they provide additional introduction on the relevance of CD3/46 activation model. They also demonstrate that atorvastatin treatment does not interfere with proliferation and viability of their cells in this system. They also show that atorvastatin treatment does not alter total cellular cholesterol levels.

There was a request by this reviewer for additional mechanistic studies linking cholesterol homeostasis with IL-10 production. This reviewer requested a mouse model of disease. The revised manuscripts now provide some correlative evidence linking IL-10 with cholesterol via control of c-MAF expression. This is a modest advance in mechanism, and I am not sure that it would have satisfied the reviewer’s concerns. I defer to the editor on this issue.

All other minor concerns appear to be addressed.

Thank you for these comments.

Reviewer #4 (Remarks to the Author):

This is a very novel paper concerning a major medical issue. However, I will comment only on the statistical issues.

Specific Comments:

1. Line 365: IQR= (2,44) is too wide, which tells that the variability is too much. This implies that the sample sizes are not large enough.

Our analysis described here relates to a highly novel and innovative exploratory study in which individuals at high risk of developing rheumatoid arthritis are enrolled. All study subjects, none of whom have clinically swollen joints, undergo a synovial biopsy via arthroscopic procedure. It is known that approximately 30-40% of these individuals will go on to develop disease within 2 years. In this study, all individuals were observed over time to document the development of joint swelling (i.e. new onset RA), which, in effect, is the primary clinical outcome. This permits identification of factors defined at baseline (such as gene expression signatures in synovial tissue) that are predictors of disease progression. This has become a widely accepted clinical model for studying basic mechanisms that underpin the evolution of autoimmune arthritis, pioneered by Dr Lisa van Baarsen and her colleagues in Amsterdam. From a statistical standpoint the wide inter-quartile ranges merely reflect the fact that some individuals go on to develop arthritis very soon after the first visit, while others develop arthritis after much longer periods eg 2 years or more. The long period of follow up gives us confidence of the stratification of phenotypes. We should highlight the fact that this is a very challenging study, with limited numbers of relatively healthy subjects willing to consent to a synovial biopsy, hence recruitment is extremely difficult. Despite this, we were able to acquire 7/13 individuals who did not develop arthritis despite a follow up of at least 2 years, and 6 individuals who did progress, permitting a balanced comparison of synovial gene expression between the two groups (“progressors” versus “non-progressors”).

We have amended the text to clarify these points (see page 11).

2. Line 371: *Confidence Interval, CI should be written as (1.0, 27.9) which is also very wide.*

We have amended the text as suggested.

3. Lines 645 to 653: *The methodology is described well but for the nonparametric test with $n < 8$, I wonder what will be the power of the test? The authors should include a power analysis.*

To address the reviewer’s specific point, we have performed “*post-hoc*” power calculations using data from the manuscript where $n < 8$. Thus, in Fig 3d, where for non-treated samples IL-10 production is $2.6 \pm 1.11\%$, we calculated a power of 86.6% for cells treated with $2\mu\text{g/ml}$ of U18666A (Mean study group =1.21; $\alpha=0.05$). In Fig. 4e for $\text{IFN}\gamma^+\text{IL-10}^+$ (central panel), IL-10 production by non-treated samples is $2.37 \pm 0.71\%$ and the “*post-hoc*” power calculated for cells under lipid free media condition is 99.8% (Mean study group =1.03; $\alpha=0.05$) while for cells treated with 25-HC the power is close to 100% (Mean study group =0.40; $\alpha=0.05$). For the $\text{IFN}\gamma^+\text{IL-10}^+$ data (left panel), IL-10 production by non-treated samples is $1.76 \pm 0.82\%$ and the “*post-hoc*” power calculated for cells under lipid free media condition is 68.9% (Mean study group =1.00; $\alpha=0.05$) while for cells treated with 25-HC the value is 99.5% (Mean study group =0.36; $\alpha=0.05$).

We hope that the reviewer is reassured on this point.

4. Line 669: *What is the justification of choosing three clusters.*

This figure was generated in response to a suggestion originally proposed by Reviewer 1, commenting our first submission. Specifically, the reviewer requested that we “*show a 3 way supervised clustering of the $\text{IL-10}^+\text{IFN}\gamma^+$, $\text{IL-10}^-\text{IFN}\gamma^+$, and $\text{IL-10}^+\text{IFN}\gamma^-$ ”*. The reasoning behind this request was that we did not find significant gene expression differences between the $\text{IL-10}^+\text{IFN}\gamma^+$, $\text{IL-10}^-\text{IFN}\gamma^+$, and $\text{IL-10}^+\text{IFN}\gamma^-$ cell subsets, while the correlation between gene expression and IL-10 signal, as defined by median fluorescence intensity, identified 111 genes. The PCA plot indicates that samples do not cluster

according to their cytokine profile when data are modelled with 3 clusters, since the populations identified by cytokine expression are present in all 3 clusters. We hope that this clarifies the issue.

5. Line 676: Authors should also report FDR (False Discovery Rate) as well as FP (False Positive) and FN (False Negative).

The FDR of each individual probe is shown in Supplementary table 1. We are controlling for a FDR of 5%, but instead of using % error, we show adjusted p-values. Our gene array experiments were analysed for correlations between IL-10 expression and gene expression levels, and so were not designed with prediction modelling in mind. For this reason, we did not apply False Positive and False Negative values.

General Comments:

There are too much emphasizes are given on p-values. In view of the prevalent misuses of and misconceptions concerning p-values, some statisticians prefer to supplement or even replace p-values with other approaches. These include methods that emphasize estimation over testing, such as credibility, or prediction intervals; Bayesian methods; alternative measures of evidence, such as likelihood ratios or Bayes Factors; and other approaches such as decision-theoretic modeling and false discovery rates. I suggest authors to add few other measures.

At the editor's recommendation we have confined our responses to addressing the statistical issues above.